# Therapeutic targeting of the PLK1-PRC1-axis triggers cell death in genomically silent childhood cancer

Jing Li[1,2,3], Shunya Ohmura[1,2,3], Aruna Marchetto[1], Martin F. Orth[1], Roland Imle [3,4,5,6], Marlene Dallmayer[1,7], Julian Musa[1,2,3,8], Maximilian M. L. Knott [1], Tilman L. B. Hölting [1], Stefanie Stein[1], Cornelius M. Funk [1,2,3], Ana Sastre[9], Javier Alonso [10,11], Felix Bestvater[12], Merve Kasan[1], Laura Romero-Pérez [1,2,3], Wolfgang Hartmann [13], Andreas Ranft[14,15], Ana Banito[3,4], Uta Dirksen [14,15], Thomas Kirchner[1,16], Florencia Cidre-Aranaz [1,2,3,18] & Thomas G. P. Grünewald [1,2,3,17,18 ✉]

Chromosomal instability (CIN) is a hallmark of cancer[1]. Yet, many childhood cancers, such as Ewing sarcoma (EwS), feature remarkably 'silent' genomes with minimal CIN[2]. Here, we show in the EwS model how uncoupling of mitosis and cytokinesis via targeting protein regulator of cytokinesis 1 (PRC1) or its activating polo-like kinase 1 (PLK1) can be employed to induce fatal genomic instability and tumor regression. We find that the EwS-specific oncogenic transcription factor EWSR1-FLI1 hijacks *PRC1*, which physiologically safeguards controlled cell division, through binding to a proximal enhancer-like GGAA-microsatellite, thereby promoting tumor growth and poor clinical outcome. Via integration of transcriptome-profiling and functional in vitro and in vivo experiments including CRISPR-mediated enhancer editing, we discover that high PRC1 expression creates a therapeutic vulnerability toward PLK1 inhibition that can repress even chemo-resistant EwS cells by triggering mitotic catastrophe. Collectively, our results exemplify how aberrant *PRC1* activation by a dominant oncogene can confer malignancy but provide opportunities for targeted therapy, and identify PRC1 expression as an important determinant to predict the efficacy of PLK1 inhibitors being used in clinical trials.

[1] Max-Eder Research Group for Pediatric Sarcoma Biology, Institute of Pathology, Faculty of Medicine, LMU Munich, Munich, Germany. [2] Division of Translational Pediatric Sarcoma Research, German Cancer Research Center (DKFZ), German Cancer Consortium (DKTK), Heidelberg, Germany. [3] Hopp Children's Cancer Center (KiTZ), Heidelberg, Germany. [4] Soft tissue sarcoma Junior Research Group, German Cancer Research Center (DKFZ), German Cancer Consortium (DKTK), Heidelberg, Germany. [5] Faculty of Biosciences, Heidelberg University, Heidelberg, Germany. [6] Division of Pediatric Surgery, Department of General, Visceral and Transplantation Surgery, Heidelberg University Hospital, Heidelberg, Germany. [7] Department of General Pediatrics, University Hospital Münster, Münster, Germany. [8] Department of General, Visceral and Transplantation Surgery, Heidelberg University Hospital, Heidelberg, Germany. [9] Unidad Hemato-oncología Pediátrica, Hospital Infantil Universitario La Paz, Madrid, Spain. [10] Pediatric Solid Tumour Laboratory, Institute of Rare Diseases Research (IIER), Instituto de Salud Carlos III, Madrid, Spain. [11] Centro de Investigación Biomédica en Red de Enfermedades Raras, Instituto de Salud Carlos III (CB06/07/1009; CIBERER-ISCIII), Madrid, Spain. [12] Light Microscopy Facility, German Cancer Research Center (DKFZ), Heidelberg, Germany. [13] Division of Translational Pathology, Gerhard-Domagk-Institute for Pathology, University Hospital Münster, Münster, Germany. [14] Pediatrics III, AYA Unit, West German Cancer Centre, University Hospital Essen, Essen, Germany. [15] German Cancer Consortium (DKTK), partner site Essen, Essen, Germany. [16] German Cancer Consortium (DKTK), partner site Munich, Munich, Germany. [17] Institute of Pathology, Heidelberg University Hospital, Heidelberg, Germany. [18]These authors jointly supervised this work: Florencia Cidre-Aranaz, Thomas G. P. Grünewald. ✉email: t.gruenewald@kitz-heidelberg.de

Chromosomal instability (CIN) refers to the ongoing acquisition of genomic alterations comprising circumscribed structural aberrations, deletions of chromosome arms, translocations, and gains/losses of whole chromosomes[3,4]. Through moderate CIN, most cancer types acquire somatic copy number alterations during each cell division creating intra-tumor heterogeneity that enhances the overall 'fitness' of the cancer cell population[3,4]. However, excessive CIN causes non-viable karyotypes[3–5]. While CIN-induction in cancers being intrinsically genomically unstable may have little therapeutic benefit, it may cause massive cell death in cancers with 'silent' or nearly diploid genomes, such as pediatric cancers including Ewing sarcoma (EwS)[3,6]. Indeed, EwS genomes are remarkably silent with only a few chromosomal gains/losses, but a dominant oncogenic driver generated through fusions of *EWSR1* with variable members of the ETS family of transcription factors (in 85% *FLI1*)[7–10]. Thus, we hypothesized that CIN-induction, e.g., by uncoupling mitosis and cytokinesis[11], may constitute a new therapeutic option for EwS.

## Results and discussion

To identify a candidate target offering a large therapeutic window, we analyzed curated expression data of 929 normal tissue samples (71 tissue types) and 50 primary EwS tumors[12] for cytokinesis-related genes being highly overexpressed in EwS. As shown in Supplementary Fig. 1, protein regulator of cytokinesis 1 (*PRC1*) exhibited the greatest fold change among 77 significantly overexpressed cytokinesis-related genes (FDR-adjusted $P < 0.05$), being on average ~8-fold higher expressed in EwS than in normal tissues (Fig. 1a). PRC1 plays pivotal roles in orchestrating cytokinesis through bundling of antiparallel microtubules and formation of the spindle midzone via recruitment of regulator and effector proteins, such as centralspindlin, the chromosomal passenger complex, kinesins, and polo-like kinase 1 (PLK1)[11].

Exploration of a cohort of 196 EwS patients[13] uncovered that high *PRC1* expression significantly correlated with poor overall survival ($P = 0.005$), which was validated in an independent cohort of 144 patients at the protein level ($P = 0.0037$) (Fig. 1b, c). Multivariate regression analysis of data from the 96 of these 144 patients with complete clinical annotation revealed that besides metastatic disease at diagnosis, high PRC1 expression was an independent risk factor (HR = 3.1; $P = 0.04$) (Supplementary Data 1).

To test for a potential EWSR1-ETS-dependency of PRC1 in EwS, we carried out time-course knockdown experiments in A673 EwS cells harboring a doxycycline (Dox)-inducible shRNA against *EWSR1-FLI1* (A673/TR/shEF1). Indeed, knockdown of *EWSR1-FLI1* was associated with a striking reduction of *PRC1* transcription, which was confirmed in vivo in xenografted A673/TR/shEF1 cells (Fig. 2a, b). Such effect was not observed in xenografts expressing an inducible non-targeting control shRNA (A673/TR/shCtrl) (Fig. 2b).

Since EWSR1-ETS oncoproteins regulate many of their target genes through DNA-binding at enhancer-like GGAA-microsatellites (mSats)[14–18], we analyzed available DNase-Seq and ChIP-Seq data from EwS cell lines (A673, SK-N-MC)[15,19] for EWSR1-FLI1 and histone marks (H3K4me1, H3K27ac). We found a prominent EWSR1-FLI1 signal located ~90 kb telomeric from the *PRC1* promoter that mapped to a GGAA-mSat within open chromatin (DNase 1 hypersensitivity site) (Fig. 2c). In both cell lines, silencing of *EWSR1-FLI1* was accompanied by a loss of EWSR1-FLI1 signals at this locus, and, more interestingly, with a loss of H3K27 acetylation usually marking active enhancers (Fig. 2c). The EWSR1-FLI1-dependent enhancer activity of this GGAA-mSat was confirmed in reporter assays in A673/TR/shEF1 cells with/without silencing of *EWSR1-FLI1* (Fig. 2d). Since GGAA-mSats are typically polymorphic, we tested several cloned fragments (1053 bp including flanking regions) from three

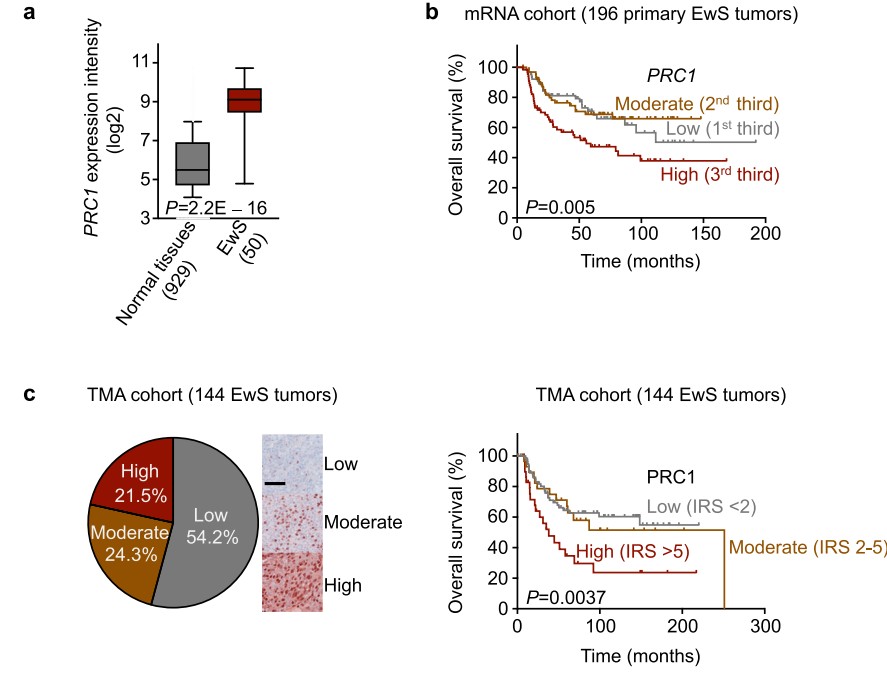

**Fig. 1 PRC1 is overexpressed in EwS and correlates with poor overall survival. a** Analysis of *PRC1* mRNA expression in 50 EwS and 929 normal tissues (comprising 71 normal tissue types). Data are represented as box plots. Horizontal bars indicate median expression levels, boxes the interquartile range, and whiskers the 10th and 90th percentile. Two-sided Mantel-Haenszel test. **b** Kaplan-Meier survival analysis of 196 EwS patients stratified by thirds of *PRC1* mRNA expression (Low, Moderate, High). Two-sided Mantel-Haenszel test. **c** Left: IHC staining of a TMA comprising 144 EwS tumors for PRC1. Scale bar = 100 μm. Right: Kaplan-Meier analysis of overall survival of 144 EwS patients stratified by their intra-tumoral PRC1 expression levels (Low: IRS < 2; Moderate: IRS 2–5; High: IRS > 5). Two-sided Mantel-Haenszel test.

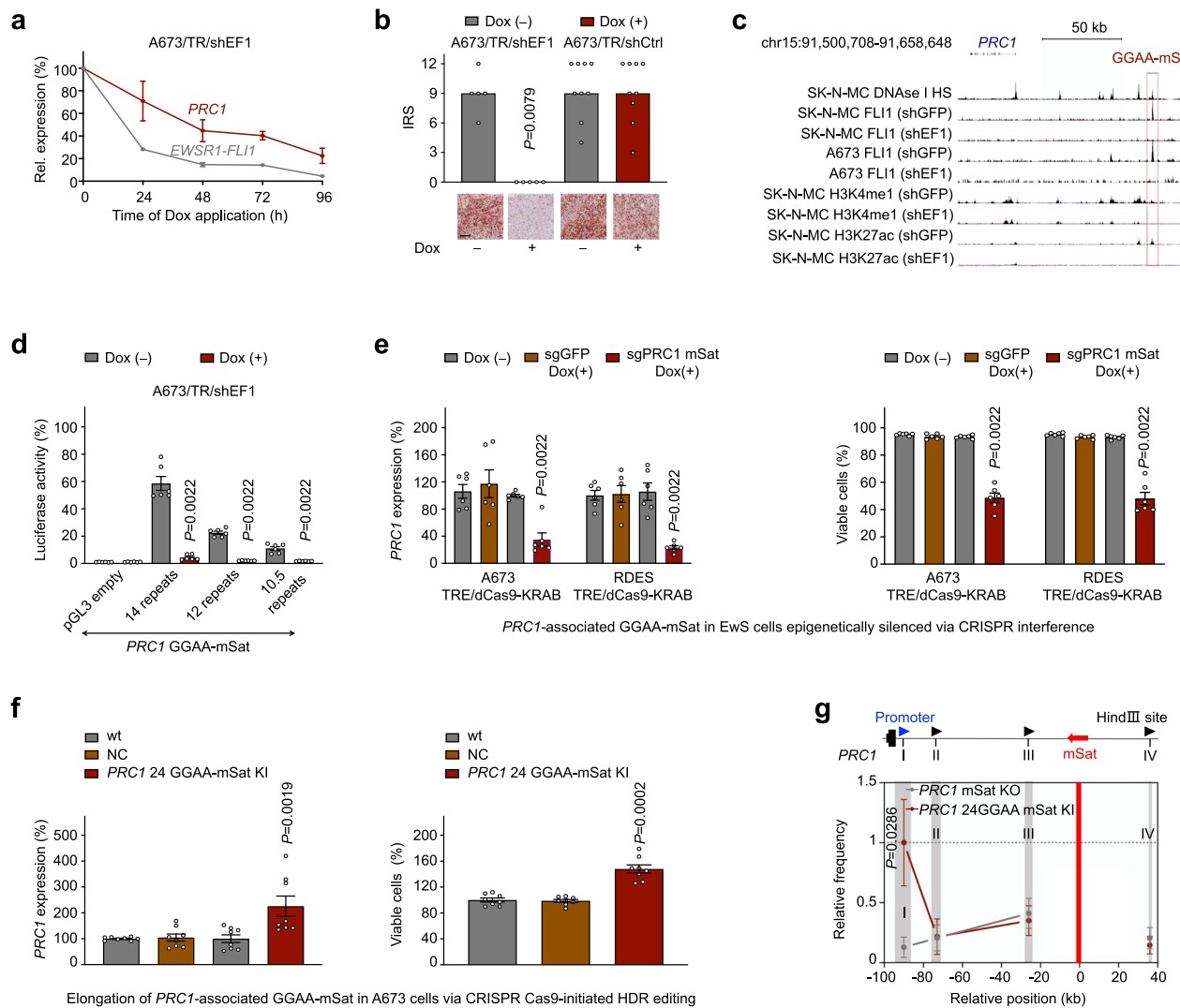

**Fig. 2 PRC1 is a direct EWSR1-FLI1 target gene. a** Time-course knockdown of *EWSR1-FLI1* in A673/TR/shEF1 EwS cells harboring a Dox-inducible shRNA against *EWSR1-FLI1* and analysis of *PRC1* and *EWSR1-FLI1* expression by qRT-PCR in vitro. Dots represent means and whiskers SEM, $n = 4$ biologically independent experiments. **b** Analysis of PRC1 expression by IHC in xenografts derived from A673/TR/shEF1 and A673/TR/shCtrl cells with/without Dox-treatment for 96 h in vivo. Data are displayed as individual dots. Horizontal bars represent the median, *n* indicates the number of biologically independent experiments/cell line: A673/TR/shCtrl $n = 9$, A673/TR/shEF1 $n = 5$; scale bar = 200 μm. **c** Integrative genomics view (hg19) of the *PRC1* locus from data of A673 and SK-N-MC cells being transfected with shRNAs targeting either *GFP* (shGFP; negative control) or *EWSR1-FLI1* (shEF1). **d** Luciferase reporter assays in A673/TR/shEF1 cells with/without knockdown of *EWSR1-FLI1* (Dox+/−) 72 h after transfection. Data are displayed as individual dots. Horizontal bars represent means and whiskers SEM, $n = 6$ biologically independent experiments. **e** Analysis of *PRC1* expression by qRT-PCR (left) and viable cells (right) in TRE-regulated dCas9-KRAB A673 and RDES cells transduced with sgRNAs against the *PRC1*-associated GGAA-mSat and *GFP* sgRNAs (negative control) 15d after addition of Dox (2 μg/ml). Data are displayed as individual dots. Horizontal bars represent means and whiskers SEM, $n = 6$ biologically independent experiments. **f** Analysis of *PRC1* expression by qRT-PCR (left) and viable cells (right) in wildtype (wt), CRISPR Cas9-edited negative control (NC), and CRISPR Cas9-initiated HDR edited (insertion of 24-GGAA-repeats) A673 cells 72 h after seeding. Data are displayed as individual dots. Horizontal bars represent means and whiskers SEM, $n = 8$ biologically independent experiments. **g** Relative crosslinking frequencies observed in CRISPR Cas9-initiated HDR edited *PRC1*-associated GGAA-mSat knockout (KO) A673 EwS cells are shown in gray and insertion of 24-GGAA-repeats in dark red. Gray shading indicates the position and size of the analyzed fragments, and red shading the 'vantage point' fragment harboring the *PRC1*-associated GGAA-mSat. The highest crosslinking frequency value in the graph was normalized and set to 1 in each replicate. Data are mean and SEM, $n = 4$ biologically independent experiments. Two-sided Mann-Whitney test in all panels.

different EwS cell lines that had different numbers of consecutive GGAA-repeats at this locus (A673, on average 14 repeats [14/14]; EW1, on average 12 repeats [13/11]; TC71, on average 10.5 repeats [11/10]). We noted a strong gain of enhancer activity with increasing numbers of consecutive GGAA-repeats (Fig. 2d), which has been previously described for other GGAA-mSats to contribute to inter-tumor heterogeneity in EwS[17,18]. Another genetic variation in the cloned fragments was excluded by whole-genome sequencing and focal Sanger-sequencing. Notably, the

average number of GGAA-repeats at this mSat corresponded to the PRC1 expression levels across EwS cell lines at both the mRNA and protein levels (Supplementary Fig. 2b). Consistently, analysis of publicly available proteomics data (https://depmap.org/portal/) comprising 4 EwS cell lines including TC71 and A673 showed that TC71 exhibited the lowest and A673 the highest PRC1 protein levels (Supplementary Fig. 2c).

To confirm the regulatory effect of this GGAA-mSat on *PRC1* transcription, we carried out clustered regulatory interspaced

short palindromic repeats interference (CRISPRi) experiments in two EwS cell lines (A673, RDES). Epigenetic interference with this GGAA-mSat markedly reduced PRC1 expression and proliferation of derivative EwS cells (A673/TRE/dCas9/KRAB; RDES/TRE/dCas9/KRAB) (Fig. 2e). The relationship between PRC1 and this GGAA-mSat was further probed by CRISPR Cas9-initiated homologous deficiency repair (HDR) DNA editing. To avoid knocking out PRC1 itself, which may have led to unphysiologically low or absent PRC1 expression levels, we chose to knockout (KO) its associated enhancer-like GGAA-mSat. Similar to CRISPRi, the genetic knockout (KO) of the PRC1-associated GGAA-mSat was accompanied by significantly lower PRC1 expression levels, proliferation, and sphere-formation in both cell lines (Supplementary Fig. 2d–h). Conversely, replacement of the wildtype mSat comprising 14 consecutive GGAA-repeats by a 'longer' haplotype (24 GGAA-repeats) significantly increased PRC1 expression, proliferation, and sphere-formation (Fig. 2f, Supplementary Fig. 2g, h). To elucidate whether PRC1 was directly regulated by EWSR1-FLI1, we performed chromosome conformation capture (3C)-PCR assays in A673 cells with either a KO or genetically elongated PRC1-associated GGAA-mSat. These experiments confirmed a physical interaction of both DNA elements, which was abrogated by KO of the PRC1-associated GGAA-mSat (Fig. 2g, Supplementary Fig. 2i). In synopsis, our results indicated that PRC1 is a direct EWSR1-FLI1 target whose high but variable expression is controlled by EWSR1-FLI1-binding to a polymorphic enhancer-like GGAA-mSat.

To explore the functional role of PRC1 in EwS, we generated three EwS cell lines (RDES, SK-N-MC, TC32) with Dox-inducible shRNAs against PRC1 (shCDS/shUTR) and non-targeting controls (shCtrl). Knockdown efficacy was confirmed by qRT-PCR and western blotting (Supplementary Fig. 3a, b). We first carried out transcriptome profiling of two of these EwS cell lines (RDES, SK-N-MC) after Dox-induced PRC1 silencing (Supplementary Data 2). Functional gene-set enrichment followed by weighted correlation network analysis revealed that PRC1 had pleiotropic effects on diverse cellular functions linked amongst others to DNA packaging, chromosome formation, cell morphology, and growth (Fig. 3a, Supplementary Fig. 3c). In line with these predictions, we noted that Dox-induced PRC1 silencing strongly reduced cell proliferation while inducing apoptosis in all three cell lines (Fig. 3b, c). Similarly, PRC1 knockdown decreased clonogenic capacity and anchorage-independent growth in vitro and strongly inhibited tumor growth in vivo (Fig. 3d, e, Supplementary Fig. 3d, e). Immunohistological assessment of the xenografts confirmed that PRC1 was efficiently downregulated, which was associated with a significantly reduced Ki-67 positivity from ~80% down to ~30%, but ~8-fold higher rates of apoptosis (cleaved caspase-3) (Fig. 3f). Notably, although PRC1 knockdown appeared to strongly reduce tumor growth, the remaining proliferation rate of the tumor cells was still relatively high compared to other malignancies, such as breast cancer (for review see ref. [20]), suggesting that the massive induction of apoptosis rather than the mere blockage of mitotic activity was the main driver of reduced tumor growth.

Based on PRC1's essential function in spindle midzone formation[11], we reasoned that these phenotypes may be explained by excessive CIN resulting from cytokinesis defects upon PRC1 knockdown. Indeed, cell cycle analysis of synchronized EwS cells showed that PRC1 silencing led to a higher fraction of cells in the G2/M phase over time, which is indicative of a delayed transition through the G2/M-phase that may contribute to the generation of tetraploid cells (Supplementary Fig. 3f). Fluorescence-in-situ-hybridization (FISH) analyses using pan-centromere probes revealed that PRC1 knockdown for only 48 h induced likely non-viable chromosomal abnormalities in ~70% of EwS cells

compared to ~10% in controls (Fig. 3g). Moreover, time-lapse live-cell imaging of TC32 and RDES EwS cells with/without PRC1 knockdown demonstrated that PRC1 silencing was associated with a significantly higher number of cells exhibiting mitotic activity without subsequent cytokinesis resulting in chromosome missegregation in both cell lines ($P = 0.0022$) (Fig. 3h, Supplementary Fig. 3g). These in vitro observations corresponded to a higher degree of nuclear pleomorphism and the presence of so-called, 'monster' cells with bizarre, aneuploid, and often multi-lobulated nuclei, as well as to higher rates of DNA double-strand breaks as indicated by γ-H2AX stains in vivo (Fig. 3i, j). The human origin of monster cells was validated by immunohistochemistry (IHC) using a human-specific anti-mitochondrial antibody (Fig. 3i). In sum, these results indicated that PRC1 is essential for proper cell division in EwS and that disruption of the delicate balance between mitosis and cytokinesis causes non-viable karyotypes in this otherwise genetically silent pediatric cancer.

Based on these results, we hypothesized that targeting cytokinesis may offer a new therapeutic strategy for EwS. Since––to the best of our knowledge––there are no direct inhibitors against PRC1 available, we explored whether targeting its major upstream interacting partner PLK1 could serve as an alternative target. Indeed, PLK1 phosphorylates and binds to PRC1, which enables the formation of the PLK1-PRC1 complex that is critical for its translocation to the central spindle to initiate cytokinesis[11,21,22]. In accordance, both genes were highly significantly co-expressed in patient EwS tumors ($n = 196$, $r_{Pearson} = 0.58$, $P = 2.2^{-16}$) (Supplementary Fig. 4a), and combining both markers yielded a highly significant association with worse overall survival ($P = 0.0013$) (Supplementary Fig. 4b, c). Thus, we treated our EwS models with two small-molecule ATP-competitive PLK1 inhibitors (BI2536, BI6727 [alias Volasertib]) that are currently used in clinical trials (Supplementary Data 3). Both inhibitors showed strong anti-proliferative effects on EwS cells at lower nanomolar levels in vitro (Fig. 4a, b). Even more strikingly, direct and indirect downregulation of PRC1 either by RNA interference or genetic KO of the PRC1-associated GGAA-mSat dramatically diminished the sensitivity of EwS cells toward both PLK1 inhibitors (Fig. 4a, b). Conversely, upregulation of PRC1 via CRISPR-mediated elongation of the PRC1-associated GGAA-mSat significantly increased their sensitivity toward both inhibitors (Fig. 4b). Importantly, it should be noted that the percentage of viable cells at the time of the beginning of PLKi-treatment was ~90–95% and that the percentage of apoptotic cells was equal in the control and treatment groups (Supplementary Fig. 4d, e). Similar to long-term suppression of PRC1, long-term treatment of EwS cells with both inhibitors strongly reduced clonogenic and anchorage-independent growth, and significantly induced apoptosis (Fig. 4c, d, Supplementary Fig. 4f). Together, these findings suggested that EwS cells with high PRC1 expression are very sensitive to PLK1 inhibition and that this sensitivity can be almost abolished by the suppression of PRC1.

To confirm the PRC1-dependency of PLK1 inhibition in vivo, we used our CRISPR Cas9-initiated HDR edited A673 EwS cells and xenografted them in NOD/scid/gamma (NSG) mice. Once tumors were palpable, mice were treated with BI2536 (40 mg/kg) or BI6727 (30 mg/kg) once per week via tail vein injection at clinically achievable dosages (Supplementary Data 4). Treatment of mice xenografted with highly PRC1 expressing EwS cells led to strong inhibition of tumor growth and even tumor regression after only three cycles of treatment without overt adverse effects, such as weight loss (Fig. 4e, f, Supplementary Fig. 4g, h). In addition, both PLK1-inhibitors led to a significant increase in the number of aneuploid cells and, monster' cells (Supplementary Fig. 4i). Strikingly, this effect could be abrogated by genetic KO of

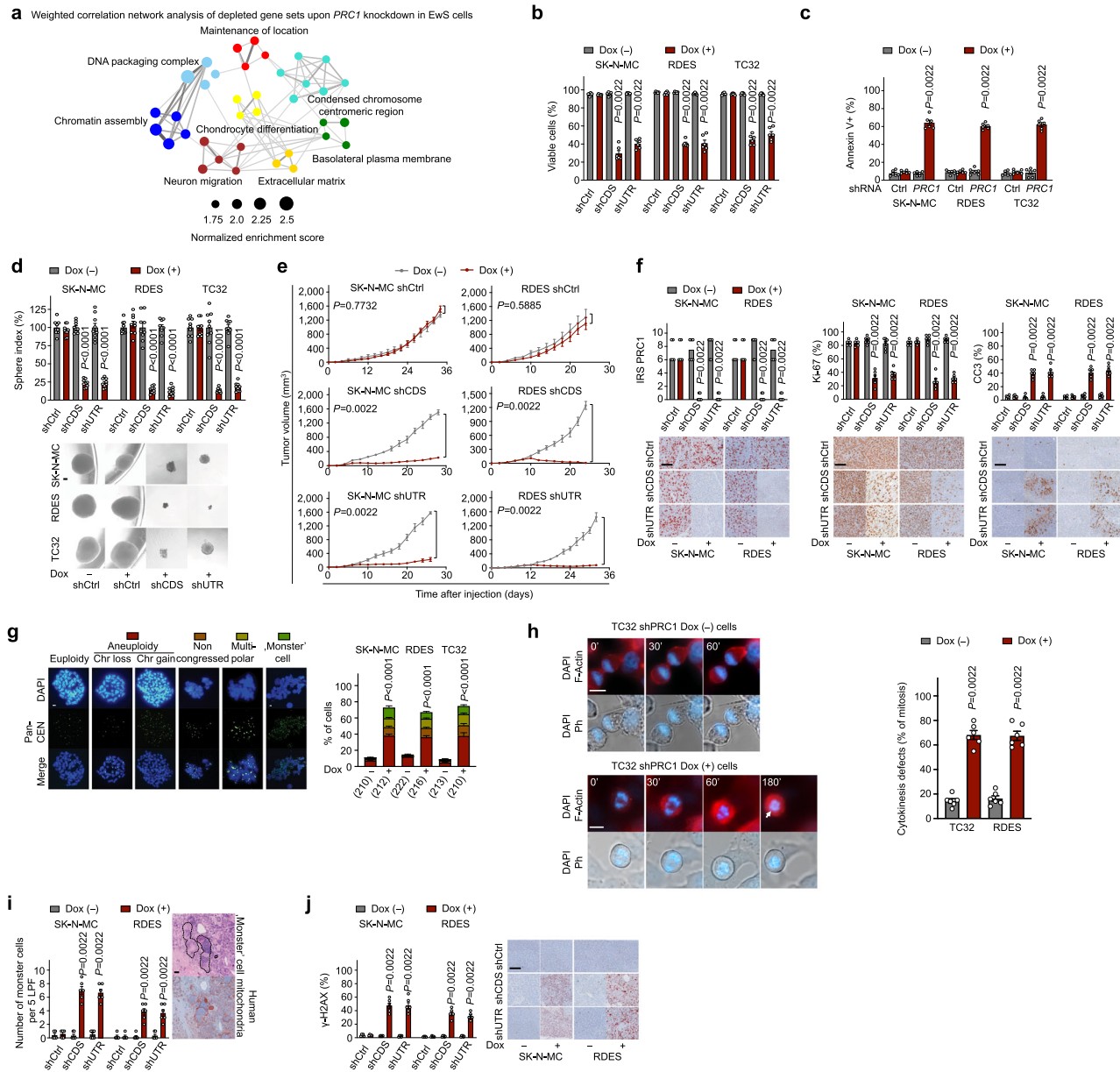

**Fig. 3 PRC1 safeguards genome stability in EwS cells. a** Transcriptomic network analysis after *PRC1* knockdown. **b** Cell viability 72 h after *PRC1* silencing. Data are displayed as individual dots. Horizontal bars represent means and whiskers SEM, *n* = 6 biologically independent experiments; two-sided Mann-Whitney test. **c** Analysis of apoptosis (summary of both shPRC1). Data are displayed as individual dots. Horizontal bars represent means and whiskers SEM, *n* = 6 biologically independent experiments; two-sided Mann-Whitney test. **d** Sphere-formation after *PRC1* knockdown. Data are displayed as individual dots. Horizontal bars represent means and whiskers SEM, *n* = 9 biologically independent experiments; two-sided Mann-Whitney test. Representative images; scale bar = 50 μm. **e** The volume of xenografts with/without *PRC1* silencing. Data are mean and SEM, *n* indicates the number of animals per condition: *n* = 5 shCtrl and *n* = 6 shCDS/shUTR in both cell lines. Two-sided Mann-Whitney test. **f** Representative micrographs and quantifications of PRC1 (IRS), Ki-67, and cleaved caspase-3 (CC3). Data are displayed as individual dots. Horizontal bars represent median IRS of PRC1 or means and whiskers SEM for positivity, respectively; the number of animals per condition is as in **e**; two-sided Mann-Whitney test; scale bar = 100 μm. **g** Representative images (scale bar = 1 μm) and quantification of aberrant metaphases. Horizontal bars represent means and whiskers SEM, *n* = 6 biologically independent experiments. Parentheses indicate a total number of analyzed mitoses (summary of both shPRC1). Two-sided Chi-squared test. **h** (Left) Time-lapse images of TC32 cells with/without *PRC1* knockdown. F-actin (red), DNA (blue). Arrows = merged nuclei. Scale bar = 10 μm. (Right) Quantification of cytokinesis failure (summary of both shPRC1). Data are displayed as individual dots. Horizontal bars represent means and whiskers SEM; *n* = 6 biologically independent experiments. **i** Representative micrographs of xenografts (H&E, human mitochondria (mt)), and quantification of, monster' cells per low-power field (LPF). Data are displayed as individual dots. Horizontal bars represent means and whiskers SEM, the number of animals per condition is as in **e**; two-sided Mann-Whitney test; scale bar = 20 μm. **j** Representative micrographs for γ-H2AX, and quantification per HPF. Data are displayed as individual dots. Horizontal bars represent means and whiskers SEM, the number of animals per condition is as in **e**; two-sided Mann-Whitney test; scale bar = 100 μm.

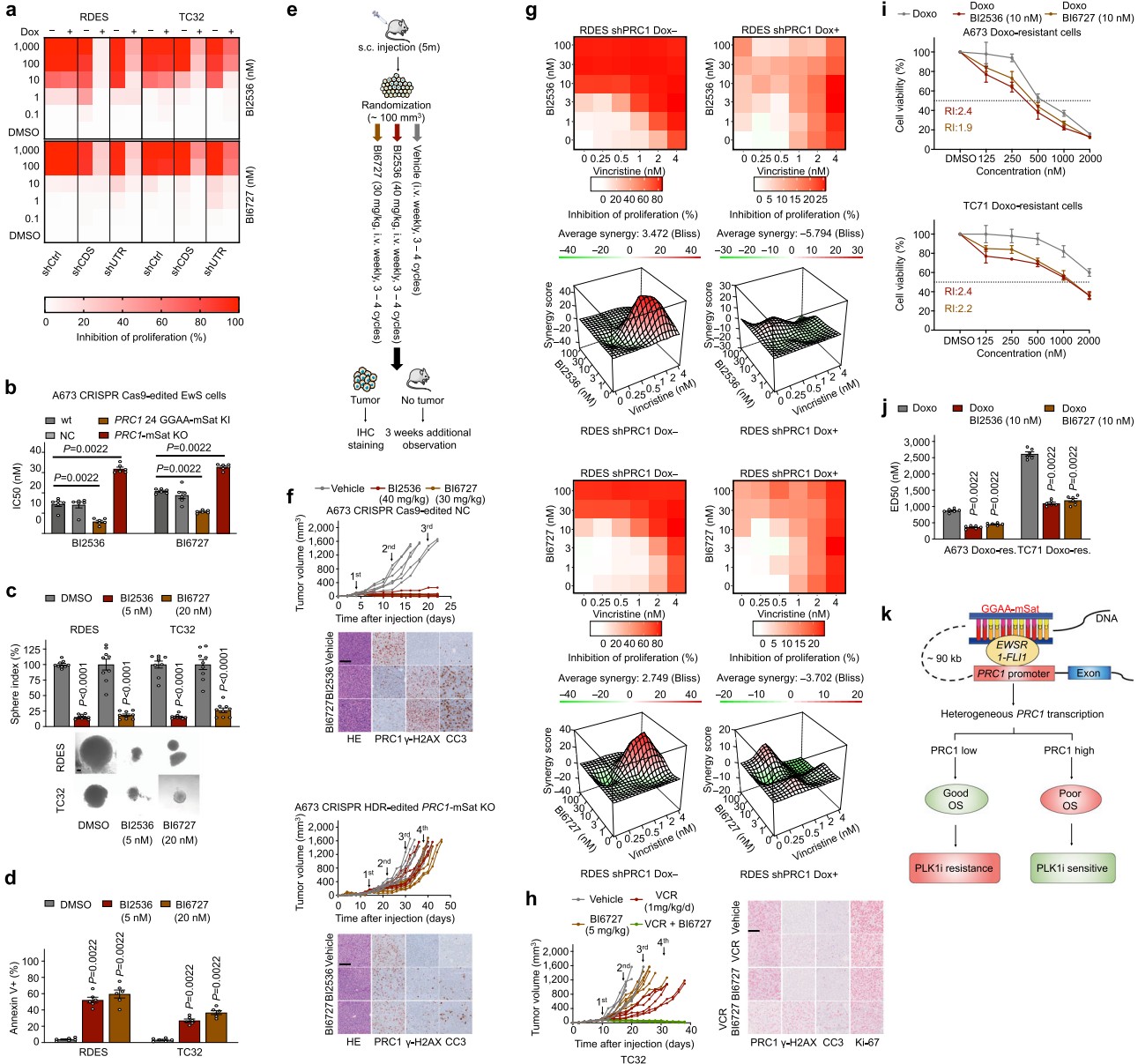

**Fig. 4 High PRC1 expression levels prime EwS cells for PLK1 inhibition. a** Heatmaps depicting the average percentage of growth inhibition in EwS cells with/without *PRC1* silencing after 72 h of PLK1 inhibition; n = 6 biologically independent experiments. **b** Analysis of IC50 in CRISPR Cas9-edited A673 cells after 72 h of PLK1 inhibition. Data are displayed as individual dots. Horizontal bars represent means and whiskers SEM, n = 6 biologically independent experiments. **c** Sphere-formation of EwS cells treated with BI2536 or BI6727. Data are displayed as individual dots. Horizontal bars represent means and whiskers SEM, n = 9 biologically independent experiments. Representative images are shown; scale bar = 50 μm. **d** Analysis of apoptosis of EwS cells with/without PLK1 inhibition for 72 h. Data are displayed as individual dots. Horizontal bars represent means and whiskers SEM, n = 6 biologically independent experiments. **e** Schematic of PLK1 inhibitor treatment in vivo. **f** Individual tumor volume curves (n = 6 per condition for NC-cells, n = 7 per condition for KO-cells). Representative micrographs of H&E and PRC1, the positivity of cleaved caspase-3 (CC3), as well as γ-H2AX per HPF. Scale bar = 100 μm. **g** Excess over Bliss analysis of RDES cells with/without *PRC1* knockdown treated with combinations of Vincristine (VCR) and the PLK1 inhibitors in vitro (red indicates synergy). Summary level (mean) data from n = 6 biologically independent experiments. **h** Analysis of tumor volume of mice treated with either vehicle, VCR (1 mg/kg/d, i.p.) alone, BI6727 (5 mg/kg, i.v.) alone, or in combination. Representative micrographs of PRC1 staining and positivity of γ-H2AX, CC3, and Ki-67 per HPF. n = 6 biologically independent animals per condition. Scale bar = 100 μm. **i** Dose-response curves of two Doxorubicin (Doxo)-resistant EwS cell lines (A673, TC71) treated with Doxo combined with BI2536 or BI6727 for 72 h. The reverse index (RI) is given. Dots represent means and whiskers SEM, n = 6 biologically independent experiments. **j** Median effective dose (ED50) for Doxo-treatment alone or in combination with PLK1 inhibitor BI2536 or BI6727 for 72 h in Doxo-resistant A673 and TC71 cells. Data are displayed as individual dots. Horizontal bars represent means and whiskers SEM, n = 6 biologically independent experiments. **k** Schematic illustrating key findings of this study. Two-sided Mann-Whitney test in all panels.

the *PRC1*-associated enhancer-like GGAA-mSat (Supplementary Fig. 4h, i). Remarkably, mice with total tumor regression did not show any sign of tumor recurrence up to 25 days after the last cycle as confirmed by necropsy and IHC (Supplementary Fig. 4j, k). In contrast, xenografts from A673 EwS cells with KO of the

*PRC1*-associated GGAA-mSat exhibited delayed tumor growth and only progressive disease during treatment despite a 4th injection was administered (Fig. 4f, Supplementary Fig. 4h, i). Together, these findings suggested that genomically silent pediatric cancers, such as EwS, may be very sensitive to PLK1

inhibition in case of high *PRC1* expression. In support of this notion, analysis of matched in vivo gene expression and drug response data from pediatric tumor types (including EwS)[23] with relatively silent genomes revealed that good responses to BI6727 (Volasertib) were observed exclusively among *PRC1* high expressing xenografts (defined by median expression; $P = 0.0325$, Fisher's exact test)—an effect not observed for *PLK1* and *MKI67* (Supplementary Data 5). However, since this dataset contained xenografts from only 4 different EwS cell lines, we extended our analyses using publicly available drug-response and gene expression data from the DepMap project, comprising 11 EwS cell lines (https://depmap.org). In this dataset, we observed a relatively strong negative correlation ($r_{Pearson} = -0.54$) of *PRC1* mRNA levels with lower cell viability upon Volasertib treatment ($P = 0.04$) (Supplementary Data 6). This correlation even remained significant when focusing on the 9 of 11 EwS cell lines that exhibited a confirmed *EWSR1-FLI1* fusion ($r_{Pearson} = -0.72$, $P = 0.02$) (Supplementary Fig. 4l). Similar to our observations made in the in vivo dataset (Supplementary Data 5)[23], such correlations were neither observed for *PLK1* nor *MKI67* regardless of the *EWSR1-FLI1* status (*PLK1*: $r_{Pearson} = -0.19/-0.4$, $P = 0.29/0.14$; *MKI67*: $r_{Pearson} = 0.04/-0.09$, $P = 0.46/0.41$) (Supplementary Data 6).

In the clinical setting, Vincristine (VCR) and Doxorubicin (Doxo) are highly active chemotherapeutics employed in first-line treatment and relapsed EwS[10]. Hence, we assessed the potential synergistic effects of PLK1 inhibition with both drugs. As shown in Fig. 4g, the microtubule-destabilizing drug VCR showed highly synergistic effects (positive Bliss score) at nanomolar concentrations in RDES cells, which was strongly diminished upon PRC1 knockdown. This PRC1-dependent effect was confirmed in TC32 EwS cells (Supplementary Fig. 5a) and corresponded to prior observations of the synergy of BI6727 with VCR in this cell line[24]. This combination therapy also significantly reduced the necessary IC50 of PLK1 inhibitors (Supplementary Fig. 5b), which may help to mitigate their potential adverse effects. It should be noted that almost identical effects on the PRC1-dependent PLK1 efficacy were observed in TC32 and RDES cells (Fig.4g, Supplementary Fig. 5a), which differ in their *STAG2* and *TP53* mutation status (TC32: *STAG2*-mut/*TP53*-wt; RDES: *STAG2*-wt/*TP53*-mut)[7,8], suggesting that both mutations have likely no impact on the efficacy of PLK1 inhibition in EwS cells, although it has been reported that PLK1 can phosphorylate STAG1/2[25,26]. Along the same lines, there was no statistically significant overlap between *STAG2*-mutated or *TP53*-mutated and *PRC1* highly expressing patient EwS tumors (Supplementary Data 7). Similar to the in vitro findings, we noted a strong synergistic effect of BI6727 and VCR in vivo, even when applying a 6-fold reduced dose of BI6727 (now 5 mg/kg) as inferred from our in vitro synergy assays (Supplementary Fig. 5c). In fact, while VCR or BI6727 (at the reduced dose) as single agents only delayed tumor growth, a combination of both drugs led to tumor regression in all mice without adverse effects, such as weight loss (Fig. 4h, Supplementary Fig. 5d, e). However, such a synergistic effect was not observed for the intercalating drug Doxo across both cell lines and PLK1 inhibitors regardless of the PRC1 levels (Supplementary Fig. 6a). Yet, it is noteworthy that both PLK1 inhibitors were still effective at nanomolar concentrations in EwS cell lines being highly resistant to Doxo (Supplementary Fig. 6b), and that they could partially restore their Doxo-sensitivity (Fig. 4i, j).

Collectively, our results demonstrate that the EWSR1-FLI1-mediated high PRC1 expression sensitizes EwS cells for PLK1 inhibition, and renders PRC1 as a promising predictive biomarker for therapies evoking cytokinesis defects and mitotic catastrophe (Fig. 4k). We anticipate that CIN-directed therapies have the potential to profoundly affect clinical outcomes,

especially in cancers with largely silent genomes, such as EwS. Still, careful biomarker-guided selection of patients is required to identify those patients who may benefit in particular from therapeutic modulation of CIN. In this regard, PLK4 inhibition was recently recognized as a therapeutic treatment option specifically for *TRIM37*-amplified neuroblastoma and breast cancer via causing centromere-dysfunction[27,28]. Notably, our data may also shed new light on why previous preclinical testing of PLK1 inhibition in non-preselected EwS models may have yielded heterogeneous results on its efficacy[23]. In fact, especially the TC71 cell line used in this screen exhibited a relatively low response toward BI6727 and the lowest *PRC1* expression levels among all four cell lines tested in vivo[23] (Supplementary Data 5). These findings correspond to the rather low *PRC1* expression levels and short *PRC1*-associated GGAA-mSat of the TC71 cell line as demonstrated in this study (Supplementary Fig. 2b). Yet, it should be noted that the interaction of PLK1 and PRC1 is complex: While PRC1 phosphorylation by PLK1 is required for the formation of the PRC1-PLK1 protein complex and its translocation to the spindle midzone, it has been reported that PLK1 can also negatively regulate PRC1 to prevent premature midzone formation before cytokinesis[29]. In turn, microtubules can stimulate PRC1 phosphorylation by PLK1, creating a potential negative feedback loop controlling PRC1 activity[29]. These facts imply that *PRC1* highly expressing cells may have adapted by increasing PLK1 to dampen the effects of high PRC1, which would be in agreement with our finding that *PRC1* and *PLK1* are significantly co-expressed in patient EwS tumors (Supplementary Fig. 4a). Accordingly, it is tempting to speculate that when PLK1 is inhibited, the PRC1 function may become deregulated and toxic to the cell. Although this is subject to future research, it is conceivable that the PRC1-related mechanism identified in our EwS model may be translatable to other cancers for which immuno-histochemical detection of high PRC1 levels could serve as a broadly available, and inexpensive predictive biomarker.

## Methods

**The provenience of cell lines and cell culture conditions.** Human cell lines were provided by the following repositories and/or sources: A673, HEK293T, and RDES cells were purchased from ATCC. SK-N-MC was provided by the German Collection of Microorganism and Cell Cultures (DSMZ). TC71 and TC32 cells were kindly provided by the Children's Oncology Group (COG). A673/TR/shEF1 cells were described previously[30]. EW1 cells were kindly provided by O. Delattre (Institut Curie Research Center, Paris, France). All cell lines were cultured in RPMI-1640 medium with stable glutamine (Biochrom) supplemented with 10% tetracycline-free fetal calf serum (FCS; Sigma-Aldrich) and 100 U/ml penicillin and 100 µg/ml streptomycin (Merck) at 37 °C with 5% $CO_2$ in a humidified atmosphere. Cell lines were routinely tested for mycoplasma contamination by nested PCR, and cell line identity was regularly verified by STR-profiling. Doxorubicin-resistant (Doxo-res) EwS cells were established through continuous culture with serially increasing Doxo concentrations starting at ~10 nM corresponding to pre-determined IC50 values. After successful adaptation to the given concentrations indicated by re-growth, cells were cultured with serially ascending Doxo concentrations by multiplying the IC50 values by factor 1.1–2.0. Doxo-res variants were maintained with ~200 nM Doxo.

**RNA extraction, reverse transcription, and quantitative real-time PCR (qRT-PCR).** Total RNA was isolated using the NucleoSpin RNA kit (Macherey-Nagel). One microgram of total RNA was reverse-transcribed using a High-Capacity cDNA Reverse Transcription Kit (Applied Biosystems). qRT-PCR reactions (final volume 20 µl) were performed using SYBR Green Mastermix (Applied Biosystems) mixed with diluted cDNA (1:10) and 0.5 µM forward and reverse primer on a BioRad CFX Connect instrument and analyzed using BioRad CFX Manager 3.1 software. Gene expression values were calculated using the $^{\Delta\Delta}$Ct method relative to the housekeeping gene *RPLP0* as an internal control. Oligonucleotides were purchased from MWG Eurofins Genomics (Ebersberg, Germany) (Supplementary Data 8). Thermal conditions for qRT-PCR: heat activation at 95 °C for 2 min, DNA denaturation at 95 °C for 10 s, annealing and elongation at 60 °C for 20 s (50 cycles), final denaturation at 95 °C for 30 s.

**Analysis of published DNase sequencing (DNase-Seq) and chromatin immunoprecipitation followed by high-throughput DNA sequencing (ChIP-Seq) data**. ENCODE SK-N-MC DNase-Seq (GSM736570) and ChIP-Seq data (GSE61944) were downloaded from the GEO, processed as previously described[16], and displayed in the UCSC genome browser. The following samples were used in this study:
ENCODE_SKNMC_hg19_DNAseHS_rep1
GSM1517546_SKNMC.shGFP96.FLI1
GSM1517555_SKNMC.shFLI196.FLI1
GSM1517547_SKNMC.shGFP96.H3K27ac
GSM1517556_SKNMC.shFLI196.H3K27ac
GSM1517569_A673.shGFP48.FLI1
GSM1517572_A673.shFLI148.FLI1
GSM1517570_A673.shGFP48.H3K27ac
GSM1517573_A673.shFLI148.H3K27ac

**Cloning of GGAA-mSats and luciferase reporter assays**. To assess the average enhancer activity of both alleles of the *PRC1*-associated GGAA-mSat in a given cell, 1053 bp fragments (hg19 coordinates: chr15:91,623,953–91,625,005) including *PRC1*-associated GGAA-mSats (hg19 coordinates: chr15:91,624,412–91,624,459) (Supplementary Fig. 2a) from three EwS cell lines (A673, EW1, TC71) were PCR-cloned upstream of the SV40 minimal promoter into the pGL3-Fluc vector (Promega, #E1761). Primer sequences are given in Supplementary Data 8. The presence of additional variants devoid of the GGAA-mSat was ruled out by whole-genome sequencing of the parental cell lines and Sanger-sequencing of the cloned fragments. A673/TR/shEF1 cells ($2 \times 10^5$ per well) were co-transfected with both alleles of the mSat-containing pGL3-Fluc vectors of a given cell line in equal mass and with the *Renilla* pGL3-Rluc vectors (Promega) (ratio 100:1) in a six-well plate with 2 ml of growth medium. Transfection medium was replaced by medium with/without Doxycycline (Dox) (1 µg/ml; Sigma-Aldrich) 4 h after transfection. After 72 h the cells were lysed and assayed with a dual luciferase assay system (Berthold). *Firefly* luciferase activity was normalized to *Renilla* luciferase activity.

**CRISPR interference**. The pLKO.1-puro U6 sgRNA BfuAI large stuffer lentiviral backbone for expression of the sgRNAs and the pHAGE TRE dCas9-KRAB lentiviral plasmid[31] encoding dCas9-KRAB were obtained from Addgene (#52628 and #50917). Candidate sgRNAs were identified by searching for $G(N)_{20}GG$ motifs 150 bases upstream/downstream of the *PRC1*-associated GGAA-mSat that correspond to the nucleotide requirements for U6 Pol III transcription and the spCas9 PAM recognition element (NGG) (Supplementary Data 8)[32]. Relevant off-target matches were determined and ruled out by CRISPR-Cas9 guide RNA design checker (IDT). SgRNAs were then cloned individually into BfuAI sites in the pLKO.1-puro U6 sgRNA BfuAI large stuffer plasmid and verified by Sanger-sequencing. Lentiviral particles were produced in HEK293T packaging cells by transfection of the lentiviral expression plasmid along with psPAX2 (Addgene #12260) and pMD2.G (Addgene #12259) packaging plasmids. 48 h post-transfection, the lentivirus-containing medium was harvested and concentrated using the Lenti-X concentrator solution (Clontech). The A673 and RDES dCas9-KRAB stable EwS cell lines were generated by infecting the cell lines with pHAGE TRE dCas9-KRAB lentivirus in combination with G418 selection (800 µg/ml). A673 and RDES cells stably expressing dCas9-KRAB were transduced with pLKO.1-sgRNA lentiviral particles and treated with puromycin (1 µg/ml) and G418 (400 µg/ml) to select and maintain stable cell lines. After selection, cells were left untreated or induced with Dox (2 µg/ml) for 14 d with regular media changes. *PRC1* expression levels were analyzed by qRT-PCR and cell viability was assessed by standardized cell counting in hemocytometers using the Trypan-blue (Sigma-Aldrich) exclusion method. A sgRNA targeting the *GFP* gene was used as the negative control.

**Homology-directed repair (HDR) using Alt-R CRISPR-Cas9 system and ultramer oligos**. We used the publicly available gRNA design tool from IDT (https://eu.idtdna.com/site/order/designtool/index/CRISPR_CUSTOM) to identify high-scoring gRNAs to two target regions flanking the *PRC1*-associated GGAA-mSat motifs (Supplementary Data 8). Each crRNA (CRISPR RNA) (IDT, Alt-R CRISPR-Cas9 crRNA, custom design) was mixed with tracrRNA (trans-activating crRNA) (IDT, Alt-R CRISPR-Cas9 tracrRNA) in equimolar concentrations to form crRNA:tracrRNA duplexes (10 µM). The upstream and downstream RNA complexes were mixed in equimolar concentrations and then incubated with Cas9 HIFI V3 nuclease (IDT, Alt-R S.p. HiFi Cas9 Nuclease V3, #369268) for 5 min at room temperature (RT) to assemble the RNP complex[33]. A negative control (ctrl) ctRNP was established at the same time (IDT, Alt-R Cas9 Negative Control). Ultramer ssODN donor of either homologous arms alone (50 bp upstream and downstream) or homologous arm plus a 24-GGAA-insert (196 bp) was added to the RNP complex individually and incubated for 5 min at RT to generate the RNP mixture (ssODN final concentration 4 µM). The RNP mixture was added to a diluted CRISPR MAX reagent (Invitrogen) to generate the RNP lipid complex of a final concentration of 90 nM. The ctRNP transfection solution was then incubated for 13 min at RT. A673 and RDES EwS cells were synchronized in the G2/M phase via incubation with Nocodazole for 16 h (200 ng/ml) followed by a 1 h release before transfection. 50 µl of the ctRNP transfection solution were

added to 100 µl synchronized cell suspension (40,000 cells) with HDR enhancer (final concentration 30 µM) (IDT, Alt-R® HDR Enhancer) seeded in triplicates in 96-well plates. Transfected cells were incubated at 37 °C (5% $CO_2$) and the medium from cells was replaced by a fresh medium without HDR enhancer after 24 h. Genomic DNA isolation and detection of editing efficacy at a bulk level were performed 48 h after transfection by assessment of the fluorescence intensity and by PCR amplification and gel electrophoresis. Finally, single-cell cloning was carried out to select successfully edited clones, which were verified by Sanger-sequencing (Supplementary Fig. 2d).

**Chromatin conformation capture (3C) PCR assays**. The 3C procedure was performed according to established protocols[34,35] with minimal modifications. Briefly, $1 \times 10^7$ cells were treated with 1% formaldehyde in phosphate-buffered saline (PBS)/10% FCS and incubated for 10 min at RT while tumbling, and then this cross-linking reaction was quenched by the addition of glycine to 0.125 M. After washing with PBS, the fixed cells were incubated for 15 min in an ice-cold lysis solution (10 mM Tris, pH 8.0, 10 mM NaCl, 0.2% NP40 and a protease inhibitor cocktail [Roche]) under constant shaking. The formaldehyde-crosslinked nuclei were harvested and washed with 1.2× RE buffer (50 mM potassium acetate, 20 mM tris acetate [pH 7.9], 10 mM magnesium acetate, 100 µg/ml bovine serum albumin [BSA]) and then re-suspended into 0.5 ml 1.2× RE buffer with the addition of sodium dodecyl sulfate (SDS) to the final concentration of 0.3%. After incubation for 20 min at 65 °C followed by 40 min at 37 °C with shaking at 1200 rpm, Triton X-100 was added to a final concentration of 2% and the solution was further incubated for 1 h at 37 °C with shaking at 1200 rpm to sequester the SDS. The DNA was first digested with 400U HindIII-HF (NEB) for 6 h at 37 °C with shaking followed by the addition of another 400U HindIII-HF for overnight digestion at 37 °C with shaking at 1200 rpm. The restriction endonuclease was inactivated by the addition of SDS to a final concentration of 1.6% and incubation for 20 min at 65 °C under shaking at 1200 rpm. After collection of the digested solution, the solution was diluted in 7 ml 1× ligation buffer (30 mM Tris-HCI [pH 7.8], 10 mM $MgCI_2$, 10 mM DTT, 1 mM ATP, and 100 µg/ml BSA). Then, Triton X-100 was added to a final concentration of 1%, and the solution was incubated at 37 °C for 1 h while gently shaking. Next, 100U of T4 DNA Ligase (HC) (Promega) was added, and the DNA was ligated for 4 h at 16 °C and then for 30 min at RT. Crosslinks were reversed by incubating the sample at 65 °C overnight in the presence of 300 µg proteinase K. A total of 300 µg RNase A was added and incubated for 45 min at 37 °C. DNA was purified by extraction with phenol, phenol-chloroform, and chloroform followed by precipitation with ethanol and dissolved in 100 µl of 10 mM Tris (pH 7.5). Digestion efficiencies were monitored by SYBR qPCR with primer pairs that amplify genomic regions containing or devoid of HindIII digestion sites (Supplementary Fig. 2i). The ligation products were analyzed by TaqMan real-time PCR (Supplementary Data 8). A random ligation control was generated using a bacterial artificial chromosome (BAC; clone CH17-26I20, CHORI BACPAC Resources Center) harboring all ligation products under study. Thirty micrograms of BAC clone were digested with HindIII-HF (NEB) and then ligated at a high DNA concentration, and extracted by phenol, phenol-chloroform, and chloroform followed by precipitation with ethanol. After digestion, cross-links were reversed, and DNA was purified. The DNA was serially diluted and used to generate a standard curve to which all 3C products were normalized. The ligation frequency was determined as a ratio of the 3C ligation product to the corresponding product in the random ligation control. For further normalization, the frequencies of ligation between fragments of the locus of *ERCC3*, a ubiquitously expressed gene, were determined[34].

**Generation of Dox-inducible short hairpin RNA (shRNA) expressing cells**. For long-term experiments, human EwS cell lines RDES, SK-N-MC, and TC32 were transduced with lentiviral pLKO-TET-ON all-in-one vector system (Plasmid #21915, Addgene) containing a puromycin resistance cassette for Dox-inducible expression of shRNAs against *PRC1* (shPRC1) or a non-targeting control shRNA (shCtrl). Dox-inducible vectors were generated according to a publicly available protocol[36] using In-Fusion HD Cloning Kit (Clontech) containing as insert a non-targeting shRNA (shCtrl), and two shRNAs against *PRC1* (Supplementary Data 8). Vectors were amplified in Stellar Competent Cells (Clontech) and integrated shRNA was verified by Sanger-sequencing (sequencing primer: 5′-GGCAGGGA-TATTCACCATTATCGTTTCAGA-3′). Lentiviral particles were generated in HEK293T cells. Virus-containing supernatant was collected to infect the human EwS cell lines. Successfully infected cells were selected with 2 µg/ml puromycin (InVivoGen). The shRNA expression for *PRC1* knockdown in EwS cells was achieved by adding 1 µg/ml Dox every 48 h to the medium. Generated cell lines were designated RDES/TR/shCtrl, RDES/TR/shCDS, RDES/TR/shUTR, SK-N-MC/TR/shCtrl, SK-N-MC/TR/shCDS, SK-N-MC/TR/shUTR, TC32/TR/shCtrl, TC32/TR/shCDS and TC32/TR/shUTR.

**Gene expression microarray analysis and functional gene-set enrichment analyses**. To identify genes involved in cytokinesis in EwS, we used curated publicly available gene expression data generated on Affymetrix HG-U133Plus2.0 DNA microarrays for 979 samples comprising 50 EwS tumor samples and 929 normal tissue samples (71 normal tissue types)[12]. First, differential gene expression

and statistical significance levels were calculated with *limma* (R package, v,3.44.3)[37]. Then, the resulting *P*-values were adjusted for multiple testing based on false discovery rate (FDR) correction. Finally, only differentially expressed genes with significant fold changes (FCs) (FDR-adjusted *P* < 0.05) were analyzed for Gene Ontology (GO)-term enrichment using clusterProfiler (R package, v,3.16.1)[38]. Annotation of GO-terms was done using the *org.Hs.eg.db* (R package, v,3.11.4).

To assess the impact of *PRC1* on gene expression and on alternative splicing in EwS, a microarray analysis was performed. To this end, $5 \times 10^5$ cells were seeded in T25 flasks and treated with Dox for 60 h (Dox-refreshment after 48 h). Thereafter, total RNA was extracted with the ReliaPrep miRNA Cell and Tissue Miniprep System (Promega) and transcriptome profiled at IMGM laboratories (Martinsried, Germany). RNA quality was assessed with a Bioanalyzer and samples with RNA integrity numbers (RIN) > 9 were hybridized to Human Affymetrix Clariom D microarrays. Data were quantile normalized with Transcriptome Analysis Console (v4.0; Thermo Fisher Scientific) using the SST-RMA algorithm. Annotation of the data was performed using the Affymetrix library for Clariom D Array (version 2, human) on the gene level. DEGs with consistent and significant FCs across shRNAs and cell lines were identified as follows: First, the normalized gene expression signal was log2 transformed. To avoid false discovery artifacts due to the detection of only minimally expressed genes, we excluded all genes with a lower or just minimally higher gene expression signal than that observed for *ERG* (mean log2 expression signal of 6.29), which is known to be virtually not expressed in *EWSR1-FLI1* positive EwS cell lines[9]. Accordingly, only genes with mean log2 expression signals (w/o Dox condition) of at least 6.5 were further analyzed. The FCs of the shCtrl samples and both specific shRNAs were calculated for each cell line separately. Then, the FCs observed in the respective shCtrl samples were subtracted from those seen in the shPRC1 samples, which yielded the FCs for each specific shRNAs in each cell line. Then these FCs were averaged to obtain the mean FC per gene across shRNAs and cell lines. To identify enriched gene sets, genes were ranked by their expression FC between the groups Dox (−/+), and a pre-ranked gene-set enrichment analysis (GSEA) was done using fgsea (R package, v,1.14.0) based on the biological processes GO definitions from MSigDB (v7.0, c2.cgp.all)[39,40]. To visualize GSEA results (Supplementary Data 9), GO terms were filtered for significance (adjusted *P* < 0.01; |Normalized enrichment score| >1.75) and used to create a symmetric GO adjacency matrix based on the Jaccard distance between GO terms and respective gene lists. GO clusters were identified using dynamicTreeCut (R package, v.1.63.1), and the GO network was visualized in Cytoscape (v.3.8.0)[41].

**Western blotting**. EwS cells with Dox-inducible shRNAs were treated with Dox (1 μg/ml for 72 h). To test for relative PRC1 protein expression levels across EwS cell lines, EwS wt cells were cultured in standard culture conditions until reaching 70% confluence. Whole cellular protein was extracted using RIPA buffer containing 1 mM $Na_3VO_4$ and protease inhibitor cocktail (Sigma-Aldrich). Proteins were separated on a 10% gel and blotted on PVDF membranes. Membranes were incubated with rabbit polyclonal anti-PRC1 antibody (1:1000, 15617-1-AP, Proteintech) or mouse monoclonal anti-GAPDH (1:800, sc-32233, Santa Cruz). Then, membranes were incubated with horseradish peroxidase (HRP) coupled anti-rabbit IgG (H + L) (1:5,000, R1364HRP, OriGene) or anti-mouse IgG (H + L) (1:3000, W402B, Promega). Proteins were detected using chemiluminescence HRP substrate (Millipore-Merck).

**Proliferation assays**. For proliferation assays, $2 \times 10^5$ EwS cells harboring Dox-inducible shRNAs were seeded in wells of 6-well plates and treated with 1 μg/ml Dox for 72 h (Dox-renewal after 48 h). Cell viability was determined by counting cells including their supernatant in standardized hemocytometers (C-Chip, Biochrom) via the Trypan-Blue (Sigma-Aldrich) exclusion method.

**Clonogenic growth assays**. For clonogenic growth assays, RDES, SK-N-MC, and TC32 cells harboring Dox-inducible shRNAs against *PRC1* and respective controls were seeded at low density (300 cells) in triplicates in 12-well plates and grown for 12–16 days with the renewal of Dox (1 μg/ml) every 48 h. RDES and TC32 EwS cells were seeded at low density (300 cells) in triplicates in 12-well plates and grown for 12–16 days with the addition of BI2536 (5 nM) or BI6727 (20 nM) or DMSO on day 1. Thereafter, colonies were stained with Crystal-Violet solution (Sigma-Aldrich), and the number and size of colonies were measured with the ImageJ Plugin *Colony area*. The clonogenicity index was calculated by multiplying the number of colonies with the corresponding colony area.

**Sphere-formation assays**. For analysis of anchorage-independent growth, the EwS cell lines RDES, SK-N-MC, and TC32 harboring shRNAs against *PRC1* and respective controls were seeded in wells at the density of 500 cells/well in 96-well Costar Ultra-low attachment plates (Corning) in 80 μl standard cell culture medium with/without Dox (1 μg/ml) and incubated for 14 days. RDES and TC32 EwS cells were seeded in wells at the density of 500 cells/well in 96-well Costar Ultra-low attachment plates (Corning) in 80 μl standard cell culture medium with addition of BI2536 (5 nM) or BI6727 (20 nM) or DMSO on day 1. The CRISPR-Cas9 edited A673 EwS cell lines and its wide type cells were seeded in wells at the density of 300 cells/well in 96-well Costar Ultra-low attachment plates (Corning) in 80 μl standard

cell culture medium. Ten microliters of fresh medium were added every 48 h. On day 14, wells were photographed, and sphere numbers and diameters were analyzed using ImageJ using the formula Area $= \pi \times A \times B/4$ (with A and B referring to orthogonal diameters).

**Cell cycle and apoptosis analysis**. For analysis of cell cycle, RDES, SK-N-MC, and TC32 cells harboring a Dox-inducible shRNA against *PRC1* and respective controls were synchronized by a double thymidine (T1850, Sigma-Aldrich) block/release[42]. Briefly, cells were blocked in G1/S with 1 mM thymidine for 18 h at 37 °C, then released into the S phase by washing 3× with pre-warmed serum-free media. A fresh complete medium was added to the cells and incubated for 10 h at 37 °C. The second round of thymidine to a final concentration of 1 mM was added and cells were cultured for another 18 h at 37 °C. Cells were in the G1/S boundary by then. After washing 3× with pre-warmed serum-free media, cells were seeded at $5 \times 10^5$ cells per T25 flask in the fresh medium with/without the addition of Dox (1 μg/ml). Dox was renewed 48 h after seeding. Cells were fixed with ice-cold 70% ethanol at each time point post releasing (24 h, 48 h, and 72 h), treated with 100 μg/ml RNAse (ThermoFisher), and stained with 50 μg/ml propidium iodide (PI) (Sigma-Aldrich). For analysis of apoptosis, RDES, SK-N-MC, and TC32 cells harboring a Dox-inducible shRNA against *PRC1* and respective controls were seeded at $8 \times 10^5$ cells per 10 cm dish with/without the addition of Dox (1 μg/ml) for 72 h. Dox was renewed 48 h after seeding. RDES and TC32 EwS cells were seeded at $8 \times 10^5$ cells per 10 cm dish with/without the addition of PLK1 inhibitor BI2536 (5 nM) or BI6727 (20 nM) or DMSO. For time-lapse apoptosis analysis, RDES and TC32 cells harboring a Dox-inducible shRNA against *PRC1* and respective controls were seeded at $8 \times 10^5$ cells per 10 cm dish with/without the addition of Dox (1 μg/ml) and analyzed at different time points after shRNA-mediated knockdown (24 h and 48 h). The CRISPR Cas9-initiated HDR edited A673 EwS cells and A673 wt cells were seeded at $1 \times 10^6$ cells per 10 cm dish and analyzed 24 h after seeding. Analysis of apoptosis has been performed at indicated time points by combined Annexin V-FITC/PI staining (BD Pharmingen FITC Annexin V Apoptosis Detection Kit II). Samples were assayed with BD Accuri C6 Cytometer (BD Biosciences). An example of a gating strategy was given in Supplementary Fig. 7.

**Fluorescence-in-situ-hybridization (FISH)**. After shRNA-induced *PRC1* knock-down for 48 h, EwS cells with ~70% confluency were incubated for 6 h with 0.2 mg/ml colcemid (Demecolcine Solution D1925 Sigma 10 μg/ml in HBSS). Meta-phase cells were harvested by trypsinizing, treated with a pre-warmed hypotonic solution (75 mM KCl) for 30 min at 37 °C, and fixed in fresh, ice-cold 3:1 methanol:acetic acid solution for 3 × 15 min. The fixed cells were kept overnight at 4 °C. Frozen slides were brought to RT on a heating plate. 20 μl of fixed cell suspension was dropped onto pre-cold clean slides and allowed to air-dry overnight. For chromosome counts, the cells were incubated with a pan-centromere probe coupled with fluorescein isothiocyanate (FITC) dye (Star*FISH, 1695-F-02, Cambio, UK) following the manufacturer's instructions with modifications. Briefly, to remove the acid, slides were passed through a graded alcohol series 70, 90, 100% for 2 min and allowed to air dry. Slides were then baked at 65 °C for 15 min and cooled to RT, and washed in acetone for 10 min and again air-dried. RNase was diluted in 2× standard saline concentration (SSC) (0.3 M sodium chloride, 0.03 M sodium citrate, pH7) to a final concentration of 100 μg/ml and added to the slides for 1 h at 37 °C in a humidified box. After washing the slides two times in 2× SSC for 5 min and two times for 5 min in 1× phosphate-buffered saline (PBS), the slides were immersed in a pepsin solution (100 ng/ml, pH2) for 3 min at 37 °C. Thereafter, slides were fixed in 4% paraformaldehyde for 4 min at 37 °C. After another washing step in 1× PBS for 5 min, the slides were dehydrated in 70, 90 and 100% ethanol at RT and allowed to air dry. The slides were denatured for 2 min in 70% formamide, 2× SSC, pH7, at 70 °C, and dehydrated immediately in ice-cold 70% ethanol for 2 min. After dehydrating in 70, 90, and 100% ethanol, the slides were air-dried for 30 min. The pan-centromere probe was denatured for 10 min at 85 °C and immediately chilled on ice. 10 μl of the probe were added to the slides. The slides were covered with a coverslip and sealed with rubber cement. Hybridization was carried out overnight at 37 °C. After removing the coverslips, the slides were washed in 2× SSC for 5 min and fixed in 50% formamide/2× SSC (pH7; 37 °C) two times for 5 min and then rinsed in 2× SSC two times for 5 min. The FITC-labeled probe was detected with 100 μl rabbit anti-FITC (1:200) in 4× SSC/0.1% Tween20 for 30 min at 37 °C in a humidified box. Afterward, slides were washed three times for 4 min in 4× SSC/0.1% Tween20. The signals were enhanced by FITC goat-anti-rabbit IgG (1:100) in 4× SSC/0.1% Tween20 for 30 min at 37 °C in a humidified box. After washing three times in 4XSSC/0.1% Tween20 for 4 min, the cells were counterstained with 4,6-diamidino-2 phenylindol/propidium iodide (DAPI) and embedded in the antifade solution. Slides were kept in dark at 4 °C. Analysis was carried out using a fluorescence microscope (Zeiss AxioVision MC 50, 100 W HBO lamp) with a triple dye filter for the detection of DAPI, propidium iodide, and FITC. Images were captured using AxioVision 4.9 software, and when necessary, signals were enhanced for optimal contrast using Adobe Photoshop 2020. Approximately 200 nuclei were evaluated in each set from at least 3 independent experiments (at least 35 nuclei). Inter-line variability of chromosome number was estimated on nuclei with a number of centromere signals different from the modal number of signals according to the composite karyotype of each cell line. The

karyotypic changes were classified and quantified according to the International Standard of Cytogenetic Nomenclature (ISCN)[43] and as previously described[44]: aneuploidy (chromosome loss/gain), non-congressed, multipolar, and, monster' cells.

**Time-lapse live-cell imaging**. TC32 and RDES EwS cells harboring a Dox-inducible shRNA against *PRC1* were seeded on coated Ibidi μ-slide 8 well (Ibidi GmbH, Germany) (chamber) slides at 300 μl/well ($6 \times 10^4$ cells) and incubated at 37 °C for 48 h with/without Dox (1 μg/ml) to allow for *PRC1* knockdown. On the day of imaging, cells were washed 3× with prewarmed FluoroBrite™ DMEM (A1896701, Life Technologies) and then stained for 30 min with 1× CellMask™ Deep Red Acting Tracking Stains (A57245, Life Technologies) in 250 μl Live Cell Imaging Solution (A14291DJ, Invitrogen). The staining solution was then carefully removed and the cells were subsequently counterstained with Hoechst 33342 Ready Flow Reagent (RF001, Invitrogen) for 15 min in the dark at 37 °C. After exposure, cells were washed 3× with Live Cell Imaging Solution at 37 °C and then imaged and analyzed in 300 μl prewarmed FluoroBrite™ DMEM (A1896701, Life Technologies) supplemented with 10% FCS, 4 mM GlutaMax (Invitrogen), and 20 mM HEPES (Invitrogen). The environment throughout imaging was controlled at 37 °C, 5% $CO_2$, and 90% humidity. Time-lapse images were acquired with an inverted Zeiss Axio Observer Z1 widefield microscope equipped with a piezoelectric focus and an AxioCam MRm grayscale CCD camera and controlled by ZEN pro software (Zeiss). Brightfield, Cy5 (AHF F36-523), and DAPI (Zeiss Filter Set 49, 488049-9901-000) were captured using a ×40 oil, 1.4 numerical aperture (NA) objective combined with 2 × 2 binning modes. Fluorescence and bright-filed images were acquired as Z-stacks (10 planes, 2 μm interval) in 2 × 2 binning modes at 15–20 random positions every 30 min for 18 h per condition. The resulting images were processed, analyzed, and colored using ZEN pro (Zeiss) software. When necessary, signals were enhanced for optimal contrast using Adobe Photoshop 2020. The percentage of mitotic cells that exited mitosis as a single cell was reported as those that fail cytokinesis.

**In vivo experiments**. NOD/Scid/gamma (NSG) mice have been housed in individually ventilated cages (IVC) under specific pathogen-free (SPF) conditions with strict dark/light cycles (darkness from 6 p.m. to 6 a.m.), an ambient temperature of 21–23 °C and a humidity of 45–65%. $5 \times 10^6$ EwS cells harboring an shRNA against *PRC1* or a non-targeting control shRNA were injected in a 1:1 mix of cells suspended in HBSS formulated with Calcium and Magnesium (ThermoFisher) in the right flank of 10–12 weeks old NSG mice. Both tumor diameters were measured every second day with a caliper and tumor volume was calculated by the formula $(L \times l^2)/2$. When the tumors reached an average volume of ~100 mm³, mice were randomized in two groups of which one henceforth was treated with 2 mg/ml BelaDox (Bela-pharm) dissolved in drinking water containing 5% sucrose (Sigma-Aldrich) to induce an in vivo knockdown, whereas the other group only received 5% sucrose (control). Once tumors in the control group exceeded a volume of 1500 mm³, all mice were sacrificed by cervical dislocation. Other humane endpoints were determined as follows: Ulcerated tumors, loss of ≥20% body weight, constant curved or crouched body posture, bloody diarrhea or rectal prolapse, abnormal breathing, severe dehydration, visible abdominal distention, obese Body Condition Scores (BCS), apathy, and self-isolation.

For in vivo experiments using PLK1 inhibitors, $5 \times 10^6$ CRISPR Cas9-initiated HDR edited *PRC1*-associated mSat KO A673 cells and CRISPR Cas9-edited negative control (NC) cells were subcutaneously injected in mice as described above. When the tumors reached an average volume of ~100 mm³, mice were randomly distributed in equal groups and henceforth treated once per week intravenously (i.v.) with 40 mg/kg BI2536, 30 mg/kg BI6727 (Volasertib), or vehicle (0.1 N HCl with 0.9% saline) for 3–4 treatment weeks. At the experimental endpoint or if humane endpoints as described above were reached before, mice were sacrificed by cervical dislocation. Then, xenograft tumors were extracted and fixed in 4%-formalin and paraffin-embedded (FFPE) for (immuno)histology.

For in vivo experiments using VCR and/or the PLK1 inhibitor BI6727 as a single agent or in combination, $5 \times 10^6$ TC32 EwS cells were subcutaneously injected in mice as described above. When the tumors reached an average volume of ~100 mm³, mice were randomly distributed in equal groups and henceforth treated with vehicle (0.1N HCl with 0.9% saline), VCR (alone i.p. [1 mg/kg/d] on days 0 and 1 of treatment), BI6727 (Volasertib; alone, i.v. [5 mg/kg] on day 0 of treatment), or VCR (i.p. [1 mg/kg/d] on days 0 and 1 of treatment) plus BI6727 (Volasertib; i.v. [5 mg/kg] on day 0 of treatment) for 4 treatment-cycles. At the experimental endpoint or if humane endpoints as described above were reached before, mice were sacrificed by cervical dislocation. Then, xenografted tumors were extracted and fixed in 4%-formalin and paraffin-embedded (FFPE) for (immuno) histology. Animal experiments were approved by the governments of Upper Bavaria and Northbaden and conducted in accordance with ARRIVE guidelines, recommendations of the European Community (86/609/EEC), and UKCCCR (guidelines for the welfare and use of animals in cancer research).

**Human samples and ethics approval**. Human FFPE tissue samples were retrieved from the archives of the Institute of Pathology of the LMU Munich (Germany) and the Gerhard-Domagk Institute of Pathology of the University of Münster (Germany) with approval of the institutional review boards. All patients gave informed consent. Tissue-microarrays (TMAs) were stained and analyzed with the approval of the ethics committee of the LMU Munich (approval no. 550-16 UE).

**Immunohistochemistry (IHC) and immunoreactivity scoring**. For IHC, 4-μm FFPE sections were cut and antigen retrieval was carried out by heat treatment with Target Retrieval Solution (S1699, Agilent Technologies). The slides were stained with either polyclonal anti-PRC1 antibody raised in rabbit (1:200, 15617-1-AP, Proteintech), monoclonal anti-γ-H2AX raised in rabbit (1:8000, ab81299, Abcam), or with monoclonal anti-Ki-67 raised in rabbit (1:200, 275R-15, Cell Marque) for 60 min at RT, followed by a monoclonal secondary horseradish peroxidase (HRP)-coupled horse-anti-rabbit antibody (ImmPRESS Reagent Kit, MP-7401, Vector Laboratories) or for 120 min at RT followed by Dako REAL™ Detection kit, Alkaline Phosphatase/RED Rabbit/Mouse (K5005, Dako). AEC-Plus (K3461, Agilent Technologies) or Substrate RED (K5005, Dako) was used as chromogen in the corresponding detection system. Samples were counterstained with hematoxylin (H-3401, Vector Laboratories and T865.1 Roth). For cleaved caspase-3 (CC3) staining, antigen retrieval was carried out by heat treatment with Target Retrieval Solution Citrate pH6 (S2369, Agilent Technologies). Slides were incubated with the polyclonal CC3 primary antibody raised in rabbit (1:100; 9661, Cell Signaling) for 60 min at RT followed by ImmPRESS Reagent Kit (MP-7401, Vector Laboratories) or for 120 min at RT followed by Dako REAL™ Detection kit, Alkaline Phosphatase/RED Rabbit/Mouse (K5005, Dako). DAB+(K3468, Agilent Technologies) or Substrate RED (K5005, Dako) was used as chromogen in the corresponding detection system and hematoxylin for counterstaining. For human mitochondrial staining, antigen retrieval was carried out by heat treatment with Pro Taqs II Antigen Enhancer (401602192, Quartett). Slides were incubated with the monoclonal anti-human mitochondria antibody raised in mouse (1:1,000; ab92824, Abcam) for 60 min at RT followed by ImmPRESS Reagent Kit (MP-7402, Vector Laboratories). AEC-Plus (K3461, Agilent Technologies) was used as the chromogen. For CD99 staining, slides were stained with monoclonal anti-human CD99 antibody raised in mouse (1:40, ab8855, Abcam) for 32 min using the ultraView detection kit in a VENTANA BenchMark system (Roche, Basel, Switzerland), and counterstained with hematoxylin.

To assess tissue integrity and for detection of mitotic defects, e.g., monster' cells, FFPE blocks of EwS xenografts were stained with hematoxylin and eosin (H&E). Evaluation of PRC1 immunoreactivity was carried out in analogy to the scoring of hormone receptor Immune Reactive Score (IRS) ranging from 0–12. This modified IRS scoring scheme has been adapted to EwS and has been described and validated previously[12,18,45–47]. The percentage of cells with expression of the given antigen was scored and classified in five grades (grade 0 = 0–19%, grade 1 = 20–39%, grade 2 = 40–59%, grade 3 = 60–79%, and grade 4 = 80–100%). In addition, the intensity of marker immunoreactivity was determined (grade 0 = none, grade 1 = low, grade 2 = moderate, and grade 3 = strong). The product of these two grades defined the final IRS. Evaluation of Ki-67, CC3, and γ-H2AX immunoreactivity was quantified based on their positive staining percentage of cells per high-power field (HPF). Final scores were determined by examination of 5–15 HPFs of at least one section per sample.

**Drug-response assays and drug combination analysis**. For PLK1 inhibitor treatment, 5000 cells/well of Dox-inducible shRNA expressing RDES and TC32 EwS cells were seeded in triplicate wells of 96-well plates. Cells were pre-treated for 48 h with Dox to induce the *PRC1* knockdown before the addition of BI2536 (S1109, Selleckchem) or BI6727 (Volasertib; S2235, Selleckchem). Cells were then treated with 0–1000 nM BI2536 or BI6727 (Volasertib) with/without Dox for an additional 72 h. The same assays were carried out with wt and CRISPR Cas9-initiated HDR edited A673 cell with a pre-incubation time of only 24 h to permit surface adherence. At the experimental endpoint, cell growth inhibition was assessed using a Resazurin assay (Sigma-Aldrich) with careful adaptations to EwS cells[48]. It should be noted that the dye Resazurin can induce fluorescence emission only in viable cells[48]. The relative IC50 concentrations were calculated using PRISM 8 (GraphPad Software Inc., CA, USA) and normalized to the respective DMSO controls.

For drug combination assays, 5000 cells of Dox-inducible shRNA expressing RDES and TC32 EwS cells were seeded in triplicate wells of 96-well plates. Cells were pre-treated for 48 h with Dox to induce the *PRC1* knockdown. Then, cells were treated with 0–100 nM PLK1 inhibitor, 0–4 nM Vincristine (VCR), and/or 0–100 nM Doxorubicin (Doxo). Inhibition of cell growth was assessed 72 h after the start of the treatment using a Resazurin assay. The excess over Bliss was calculated using *synergyfinder* (R package v.2.2.4)[49]. For assessment of the dose reduction index (DRI), 5000 cells/well of RDES and TC32 EwS cells were seeded in triplicate wells of 96-well plates 24 h before the addition of PLK1 inhibitors. Cells were treated with 0–100 nM PLK1 inhibitor, 0–4 nM VCR alone or in combination in a 1:1 constant combination ratio[50]. Inhibition of cell growth was assessed 72 h after the start of the treatment using a Resazurin assay. CompuSyn (ComboSyn, Inc.) was used for determining the DRI of each PLK1 inhibitor at the IC50 level.

For Doxo-resistance reversal assays, non-toxic concentrations (<IC10) of PLK1 inhibitors BI2536 or BI6727 in two Doxo-res EwS cell lines (A673 Doxo-res; TC71 Doxo-res) were determined by a Resazurin assay after PLK1 inhibition (0–1000 nM) for 72 h, respectively. In Doxo-resistant cell lines, the differential

in vitro efficacy of Doxo depending on the co-treatment with PLK1 inhibitor BI2536 or BI6727 was assessed by a Resazurin assay. To this end, cells were treated for 72 h with different concentrations of Doxo (0–2000 nM) and co-treated with the non-toxic dosage of PLK1 inhibitor BI2536 or BI6727 (10 nM) as defined above. The predicted ED50 concentrations were calculated using CompuSyn (ComboSyn, Inc.). For both cell lines, the reverse index (RI) was calculated by dividing the ED50 of Doxo alone by the ED50 of the combination treatment (Doxo plus 10 nM of PLK1 inhibitor).

**Statistical analyses**. If not otherwise specified, statistical data analysis was performed using PRISM 8 (GraphPad Software Inc., CA, USA) on the raw data. If not otherwise specified in the figure legends comparison of two groups in functional in vitro experiments was carried out using a two-sided Mann-Whitney test. Contingency tables of FISH counting were analyzed by the Chi-square or Fisher's exact test as appropriate. Pearson correlation analysis was done and visualized using the R package ggpubr (v,0.4.0). If not otherwise specified in the figure legends, data are presented as box-dot plots with horizontal bars representing means and whiskers representing the standard error of the mean (SEM). The sample size for all in vitro experiments was chosen empirically. For in vivo experiments, the sample size was predetermined using power calculations with $\beta = 0.8$ and $\alpha < 0.05$ based on preliminary data and in compliance with the 3R system (replacement, reduction, refinement). In retrospective Kaplan-Meier analyses of overall survival, curves were calculated from all individuals. Statistical differences between the groups were assessed by a Mantel-Haenszel test. The sample sizes of the patient cohorts were not predetermined. Retrospective analysis of survival probabilities was done on an exploratory basis. Cox multivariate statistical analyses were carried out in IBM SPSS Statistics software (version 23.0; IBM Corp). A hazard ratio (HR) > 1 indicated that patients were at risk of a worse prognosis. Potential associations of the *STAG2* and/or *TP53* mutation status with *PRC1* expression levels were assessed by a two-sided Fisher's exact test in 57 ICGC EwS cases for which the mutation status and transcriptome profiles were available[51]. To develop a prognostic index (PI), EwS patients were first stratified into two groups (i.e., *PRC1*-high and *PRC1*-low, and *PLK1*-high and *PLK1*-low expression) using the median cut-off of the expression level for each gene. The PI of every patient was calculated as the sum of each gene score, which was calculated by multiplying the expression level of a gene by its corresponding coefficient (i.e., PI = $\sum$ Cox coefficient of gene $G_i \times$ expression level of gene $G_i$). Subsequently, patients were stratified into three PI groups (i.e., low, moderate, and high) by thirds of the calculated PI as cut-offs. Kaplan-Meier analysis was performed and the difference in survival time between the lowest and highest third was assessed using a Mantel-Haenszel test. $P$-values < 0.05 were considered statistically significant. If not otherwise specified in the figure legends, all $P$-values were estimated from two-sided statistical tests.

**Reporting summary**. Further information on research design is available in the Nature Research Reporting Summary linked to this article.

## Data availability

The microarray data were deposited at the National Center for Biotechnology Information (NCBI) GEO database under the accession code GSE156559. The microarray, DNase-Seq, and ChIP-Seq data referenced during the study are available in a public repository from the GEO website (https://www.ncbi.nlm.nih.gov/geo/); accession codes: GSM736570 and GSE61944. The relative PRC1 protein expression data in EwS cell lines were extracted from Proteomics (O43663) dataset derived from DepMap portal (https://depmap.org/portal/). The *PRC1*, *PLK1*, and *MKI67* mRNA expressions of 11 EwS cell lines were extracted from the Expression 21Q public dataset and their corresponding drug sensitivities toward PLK1 inhibitor BI6727 (Volasertib) treatment were extracted from the PRSIM Repurposing Primary Screen 19Q4 derived from DepMap portal (https://depmap.org/portal/). The source data underlying Fig. 1a–c, 2a–g, 3a–j, 4a–j, and Supplementary Figs. 1, 2b–l, 3a–g, 4a–l, 5a–e, 6a, b are provided as a Source Data file. All the other data supporting the findings of this study are available within the article and its supplementary information files. Source data are provided with this paper.

## Code availability

All custom codes supporting the findings of this study are available from the corresponding author upon reasonable request.

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

## Acknowledgements

We thank Dr. Paul Northcott and Dr. Brian Gudenas for providing R scripts for weighted correlation network analyses; and Dr. Gabriel Leprivier, Dr. Veit. R. Buchholz, and Dr. Stefan M. Pfister for critical reading of the manuscript. We thank Beate Luthardt, Monika Melz, Anja Heier, and Andrea Sendelhofert for expert technical assistance. This work was mainly supported by a grant from the German Cancer Aid (DKH-70114111). In addition, the laboratory of T.G.P.G. was supported by the LMU Munich's Institutional Strategy LMUexcellent within the framework of the German Excellence Initiative, the 'Mehr LEBEN für krebskranke Kinder—Bettina-Bräu-Stiftung', the Matthias-Lackas Foundation, the Dr. Leopold and Carmen Ellinger Foundation, the Boehringer-Ingelheim Foundation, the Wilhelm Sander-Foundation (2016.167.1), the Barbara and Hubertus Trettner Foundation, the Dr. Rolf M. Schwiete Foundation, the Friedrich-Baur Foundation, the German Cancer Aid (DKH-70112257 and DKH-111886), the Gert und Susanna Mayer Foundation, the Barbara und Wilfried Mohr Foundation, the SMARCB1 association, and the Deutsche Forschungsgemeinschaft (DFG-391665916). J.L. was supported by a scholarship of the Chinese Scholarship Council (CSC), and a grant of the German Cancer Aid (DKH-70114111). M.D. was by a scholarship of the 'Deutsche Stiftung für junge Erwachsene mit Krebs', J.M. by a scholarship of the Kind-Philipp-Foundation, and C.M.F., M.K. and T.L.B.H. by scholarships from the German Cancer Aid. The laboratory of J.A. was supported by grants from the Instituto de Salud Carlos III (PI16CIII/00026; DTS18CIII/00005), Asociación Pablo Ugarte, ASION, Fundación Sonrisa de Alex, Asociación Todos somos Iván y Asociación Candela Riera. Freely available clipart used for design of parts of figures was kindly provided by Servier Medical Art (https://smart.servier.com/).

## Author contributions

J.L. and T.G.P.G. conceived the study, wrote the paper, and drafted the figures and tables. J.L. carried out in vitro and in vivo experiments. J.L. and M.F.O. performed bioinformatic and statistical analyses. A.S., J.A., U.D., A.R. and W.H. provided gene expression data or tissue microarrays. S.O., A.M., F.C.A., M.F.O., C.M.F., L.R.-P., T.L.B.H., M.M.L.K. M.D., M.K., J.M. and S.S. contributed to experimental procedures. F.B. helped in time-lapse imaging. R.I. and A.B. contributed to and coordinated animal experiments. W.H. and T.K. provided laboratory infrastructure and histological guidance. F.C.A. and T.G.P.G. provided laboratory infrastructure, coordinated project logistics, and/or supervised the study and data analysis. All authors read and approved the final manuscript.

## Funding

## Competing interests

The authors declare no competing interests.
