## [Peer Review File · Nature Communications]

Reviewers' Comments:

Reviewer #1:

Remarks to the Author:

The authors report that the protein regulator of cytokinesis 1 (PRC1) is significantly overexpressed in Ewing's Sarcoma (EWS). They report that high PRC1 expression significantly associated with poor overall survival which validated in an independent cohort of 144 patients at the protein level. They report that multivariate regression analysis that metastatic disease at diagnosis, and high PRC1 expression were independent risk-factors.

The authors show that knockdown of PRC leads to reduced cell number, apoptosis, Ki-67, anchorage-independent growth, both in vitro and in vivo. They also found a G2/M blockade with monster cells and aneuploidy with suppression of PRC1.

They speculated that targeting PRC1 upstream using inhibitors of PLK1 may represent therapy for EWS given that PLK1 phosphorylates and binds to PRC1 enabling the formation of PLK1:PRC1 complex that is critical for its translocation to the central spindle to initiate cytokinesis. They show that direct and indirect downregulation of PRC1 either by RNA interference or genetic KO of the PRC1-associated GGAA-mSat diminished the sensitivity of EWS cells toward PLK1 inhibitors.

Specific comments.

Supplementary table 1, it is unclear whether it was univariate or multivariate.

The authors should show the GGAA repeats with supplemental figure with the GGAA highlighted, otherwise hard to appreciate the number of GGAA repeats.

Figure 2d: Unclear what the 14 repeats are. What is the chromosomal location of the repeats?

Figure 2g: It is difficult to interpret the 3C results. The authors should show indicate vantage points.

It is not surprising that suppression of PRC1 leads to insensitivity to chemotherapy given that the former leads to cell cycle arrest which generally leads to reduced sensitivity to chemotherapy. Also, the observation that upregulation lead to increased sensitivity to other drugs is not surprising and it is unlikely to be specifically related to PRC1 levels or the mechanistic activity of the drug.

Does PLK1 inhibitors show a G2/M blockade with monster cells and aneuploidy in EWS?

Given the results reported here, it is disappointing that as reported by Gorlick et al that EWS do not show response to PLK1 inhibitors.

The data generated in this manuscript is sound and does demonstrate that Ewing's sarcoma is dependent on expression of PRC1 for survival, and that PRC1 is a direct target of EWSR1-FLI1, and sensitivity to PLK1 inhibitors is related to PRC1 expression. However, it is likely that the findings are specific to EWS as PRC1 is an essential gene. In addition, PLK1 inhibitors have shown disappointing single agent results in-vivo models of EWS.

Reviewer #2:

Remarks to the Author:

In this manuscript, (Therapeutic targeting of cytokinesis triggers mitotic catastrophe in genomically silent childhood cancer) Li et al. used an unbiased approach to identify cytokinesis regulatory proteins that promote Ewing Sarcoma malignancy and identified PRC1 as a potential candidate in EwS. They provide compelling evidence for EWSR1-FLI1 targeting a GGAA-mSat for PRC1 expression and establish functional relevance for this mechanism in EwS proliferation and sphere formation. Knockdown of PRC1 induced apoptosis and inhibited cell proliferation, colony formation and tumor growth. Further, PRC1 knockdown induced chromosomal aberrations

(aneuploidy) and some cells appeared as “monster cells”. Inhibition of the upstream (negative) PRC1 regulator Plk1 induced antineoplastic effects. Importantly, these effects were diminished after downregulation of PRC1 or knockout of GGAA-mSAT, indicating Plk1 inhibitors exert these effects through PRC1. Consistently, in flank xenograft experiments, both Plk1 inhibitors resulted in inhibition of tumor growth and knockout of the GGAA-mSat partially reversed these effects. In cells, vincristine (but not doxorubicin) acted synergistically with Plk1 inhibitors and synergy was lost after PRC1 knockdown.

In summary, this is a large body of work using thorough and elegant techniques that strongly suggest that EwS with elevated PRC1 may be sensitive to Plk1 inhibition, in particular when combined with vincristine. The results are quite dramatic, and the data are compelling and strongly support the main conclusions. The data are corroborated by extensive supplementary material, robust quantification and solid statistics. A major strength of this study is the fact that in vitro findings are corroborated by extensive flank tumor analysis (mainly IHC), underscoring the translational relevance of the findings reported in this paper. However, some conclusions about the mechanism/phenotypes are a bit overstated.

Major comments:

Title: the title is catchy, but this study does not provide thorough analysis of cytokinesis. Also, the term “mitotic catastrophe” is only referred to once in the manuscript (Fig. 4k, schematic) but not investigated experimentally. Therefore, using both terms (cytokinesis and mitotic catastrophe) in the title is not appropriate. Additionally, while the exact mechanism of Plk1 inhibition on PRC1 function is not investigated, the authors clearly establish the functional relevance for the Plk1-PRC1 axis, and this should be indicated in the title.

According to recent evidence, Plk1 is the major (negative) regulator of PRC1 (Hu C-K. et al. Mol Biol Cell (2012) Jul;23(14):2702). Do the authors have any evidence that expression of an unphosphorylatable PRC1 construct exerts antineoplastic effects similar to Plk1 inhibition? The authors should at least address the potential mechanism of Plk1 inhibition on PRC1 biological activity and on cytokinesis (which is mentioned in the title but not addressed in the manuscript) in the discussion.

Fig. 1B,C: The authors show PRC1 prognostic significance but later target Plk1 as the major negative upstream regulator of PRC1. Therefore, what is the prognostic value of Plk1 expression? As Supp Fig. 4A shows correlation of PRC1 and Plk1 in EwS it would be interesting to determine the multi-gene prognostic index (survival) to see whether combined expression of PRC1 and Plk1 is associated with worse prognosis.

Fig. 2B: upper panels: It is odd that the horizontal bars are all the same size around 9 Immune Reactive Score. According to the individual dots, at least the median of the third bar should be much higher than 9. Also, in the lower panels, the IHC images are of poor resolution (this is true for most IHC images throughout the manuscript).

Fig. 3: The notion of “disruption of the delicate balance between mitosis and cytokinesis” is not sufficiently corroborated. I agree the authors show chromosomal aberrations (e.g., aneuploidy), large “monster” cells and increased p-gH2AX. But this is not enough evidence to conclude these phenotypes are the result of an imbalance between mitosis and cytokinesis. Monitoring bipolar spindle formation and contractile ring formation together with chromosome segregation over time would be required in order to make such statements.

Supp Fig. 3f: The notion of higher numbers of tetraploid cells through blockage of proper G2/M transition (page 6) is not supported by the data shown. I see no evidence of tetraploid cells in Supp Fig. 3. Also, any statement about G2/M transition needs to be corroborated by analyzing cycling cells, which can be done by releasing cells from synchronization (hydroxyurea or double thymidine) and monitoring cell cycle by Flow and/or expression of cell cycle markers (e.g., cyclins) over time. As I cannot find any synchronization of cells in the figure legend or methods regarding the experiment in Supp Fig. 3, the results in Supp Fig. 3f are only a snapshot in time and no

conclusions about cell cycle transitions can be drawn. In addition, the siCtrl is missing, making it difficult to draw any conclusions.

Fig. 4g and related figures: what was the experimental readout in these experiments to assess synergy (Bliss score)? I assume these were in vitro experiments. Please clarify in figure legends.

In general, most IHC images are of very poor resolution when zoomed in. Please provide images of better quality.

I am not familiar with the term "massive genomic chaos" (page 8). Authors need to provide a definition for this term and additionally thorough scientific evidence to corroborate this statement.

Page 6, line 2: the word "proven" should be replaced (e.g., monitored, confirmed).

Minor comments:

Immune Reactive Score (IRS) is an established technique for estrogen and progesterone receptors in breast cancer, but it is not widely accepted for Ewing Sarcoma (only one Oncotarget paper from the same group). I'm not a pathologist expert but I recommend a board-certified pathologist should elaborate why this technique is superior to other pathological methodologies and that its use is justified here for Ewing Sarcoma.

For sphere formation: spheroid area was calculated using $A = \pi A \sim d^2/4$ but some spheres are not perfectly round. How was this taken into account? Was diameter taken from two different (orthogonal) measurements?

Page 8: "veil injection" should be "vein injection".

Reviewer #3:

Remarks to the Author:

This reports an intriguing and potentially important finding in the pathobiology of Ewing Sarcoma and the role of PRC1 as a novel direct target of EWSR1-FLI1 transactivation. The approach is logical and systematic, the experiments are well designed, and the data compelling. However, some broader issues or questions remain to be addressed:

1. other alterations are well established as prognostic in Ewing Sarcoma, including TP53 mutations and, especially, STAG2 loss. How does the PRC1 high subset overlap with the STAG2 loss subset? Based on Table 1 in PMID 25223734, all the Ewing lines used are TP53-mutated/STAG2-wt, except SK-N-MC which is a double mutant. This is potentially significant and possibly confounding given that PLK1 also phosphorylates STAG1/2 to enable dissociation of the cohesin complex. (PMID 31516082 and PMID 15737063)

2. although PLK1 does interact with PRC1, that interaction is not so simple. According to PMID 22621898, PLK1 negatively regulates PRC1 through phosphorylation. In turn, microtubules can stimulate PRC1 phosphorylation by PLK1, creating a potential negative feedback loop controlling PRC1 activity. The authors should comment on this aspect. For instance, perhaps PRC1-high cells have adapted by increasing PLK1 to dampen the effects of high PRC1; when PLK1 is inhibited, PRC1 function becomes totally deregulated and toxic to the cell.

3. the authors should mention and comment on the poor response of TC71 in vitro and in vivo to the same PLK Inhibitor (BI 6727) when tested by the Pediatric Preclinical Testing Program (PMID 23956067 and their ref. 22). It is not sufficient to just say "previous preclinical testing of PLK1 inhibition in non-preselected EwS models may have yielded controversial results on its efficacy" when the PRC1 status of TC71 is known.

Step-by-step responses to the Reviewers:

Authors: We thank all Reviewers for their time spent with our manuscript and their highly valuable and fair comments, which have been addressed in full.

Reviewer #1 remarks to the authors

The authors report that the protein regulator of cytokinesis 1 (PRC1) is significantly overexpressed in Ewing's sarcoma (EWS). They report that high PRC1 expression significantly associated with poor overall survival which validated in an independent cohort of 144 patients at the protein level. They report that multivariate regression analysis that metastatic disease at diagnosis, and high PRC1 expression were independent risk-factors. The authors show that knockdown of PRC1 leads to reduced cell number, apoptosis, Ki-67, anchorage-independent growth, both *in vitro* and *in vivo*. They also found a G2/M blockade with monster cells and aneuploidy with suppression of PRC1. They speculated that targeting PRC1 upstream using inhibitors of PLK1 may represent therapy for EWS given that PLK1 phosphorylates and binds to PRC1 enabling the formation of PLK1:PRC1 complex that is critical for its translocation to the central spindle to initiate cytokinesis. They show that direct and indirect downregulation of PRC1 either by RNA interference or genetic KO of the PRC1-associated GGAA-mSat diminished the sensitivity of EWS cells toward PLK1 inhibitors.

Authors: We thank this Reviewer for the concise summary of our major findings.

Specific comments:

Supplementary table 1, it is unclear whether it was univariate or multivariate.

Authors: We agree with this Reviewer and have modified the Table legend accordingly, which now reads: 'Evaluation of risk-factors of prognosis in 96 EwS patients by multivariate Cox regression analysis'.

The authors should show the GGAA repeats with supplemental figure with the GGAA highlighted, otherwise hard to appreciate the number of GGAA repeats.

Authors: We fully agree with this Reviewer and have demonstrated the number of GGAA-repeats as indexed in the reference genome (hg19; 12 GGAA repeats) in a new Supplementary Fig. 2a displaying an integrative genomics view of this locus including ChIP-Seq tracks, in which the *PRC1*-associated GGAA-mSat was highlighted in blue color. In addition, we now report the precise location of the *PRC1*-associated GGAA-mSat in the Methods section (see also response to the subsequent comment).

Figure 2d: Unclear what the 14 repeats are. What is the chromosomal location of the repeats?

Authors: We apologize for this confusion. As stated in the text of the initially submitted version of our manuscript, we cloned both alleles of the *PRCI*-associated GGAA-mSat and their flanking regions from three different Ewing sarcoma cell lines (A673, EW1, TC71). On average these cell lines had 14, 12, or 10.5 GGAA-repeats at this locus. Subsequently, we tested their combined enhancer activity by transfecting A673/TR/shEF1 cells with both alleles of a given cell line and recorded the relative luciferase signal normalized to the Renilla signal. To clarify this aspect, this procedure has now been explained in more detail in both the Methods and Results sections. In addition, we now refer to the number of repeats as the average number of repeats of both alleles per cell line, and give the precise genomic coordinates (hg19) for the *PRCI*-GGAA-mSat in the Methods section as well as in the new Supplementary Fig. 2a.

Page 4, paragraph 2: ‘Since, GGAA-mSats are typically polymorphic, we tested several cloned fragments (1,053 bp including flanking regions) from three different EwS cell lines that had different numbers of consecutive GGAA-repeats at this locus (A673, on average 14 repeats [14/14]; EW1, on average 12 repeats [13/11]; TC71, on average 10.5 repeats [11/10]). We noted a strong gain of enhancer activity with increasing average numbers of consecutive GGAA-repeats (**Fig. 2d**), which has been previously described for other GGAA-mSats to contribute to inter-tumor heterogeneity in EwS.’

Page 19, paragraph 1: **,Cloning of GGAA-mSats and luciferase reporter assays**

To assess the average enhancer activity of both alleles of the *PRCI*-associated GGAA-mSat in a given cell, 1,053 bp fragments (hg19 coordinates: chr15:91,623,953-91,625,005) including *PRCI*-associated GGAA-mSats (hg19 coordinates: chr15:91,624,412-91,624,459) (**Supplementary Fig. 2a**) from three EwS cell lines (A673, EW1, TC71) were PCR-cloned upstream of the SV40 minimal promoter into the pGL3-Fluc vector (Promega, #E1761). Primer sequences are given in **Supplementary Table 7**. The presence of additional variants devoid of the GGAA-mSat was ruled out by whole-genome sequencing of the parental cell lines and Sanger-sequencing of the cloned fragments. A673/TR/shEF1 cells (2×10^5 per well) were co-transfected with both alleles of the mSat-containing pGL3-Fluc vectors of a given cell line in equal mass and with the *Renilla* pGL3-Rluc vector (Promega) (ratio 100:1) in a six-well plate with 2 ml of growth medium. Transfection medium was replaced by medium with/without Doxycycline (Dox) (1 μ g/ml; Sigma-Aldrich) 4h after transfection. After 72h the cells were lysed and assayed with a dual luciferase assay system (Berthold). The average *Firefly* luciferase activity of both alleles was normalized to *Renilla* luciferase activity.

Figure 2g: It is difficult to interpret the 3C results. The authors should show indicate vantage points.

Authors: We thank this Reviewer for sharing his/her opinion and we apologize if the interpretation of the graph may have caused difficulties. The vantage point has been for all 3C-PCR reactions the

digested *PRC1*-associated GGAA-mSat fragment, which was highlighted in red color in our scheme above the graph. To further clarify this aspect, we highlight the vantage point with a red shading in the graph and indicate the vantage point in the corresponding figure legend. In addition, we have deleted the perhaps misleading trend lines between the tested fragments III and IV. The graphical display of our 3C-PCR data should now correspond to conventional ways of displaying such data (for comparison see Palstra *et al.* 2003 *Nature Genetics* PMID: 14517543)

It is not surprising that suppression of *PRC1* leads to insensitivity to chemotherapy given that the former leads to cell cycle arrest which generally leads to reduced sensitivity to chemotherapy. Also, the observation that upregulation leads to increased sensitivity to other drugs is not surprising and it is unlikely to be specifically related to *PRC1* levels or the mechanistic activity of the drug.

Authors: We thank this Reviewer for sharing his/her opinion. However, although silencing of *PRC1* led to strong reduction of tumor growth and massive induction of apoptosis *in vivo*, we still noted a Ki-67 positivity in around 30% of the tumor cells indicating that the tumor cells do not undergo a cell cycle arrest (Fig. 3f), which was briefly mentioned in the initial Results section. In fact, the observed remaining proliferation rate in *PRC1* knockdown conditions is even higher than for many other cancers, such as breast cancer (for review see Yerushalmi R *et al.* 2010 *Lancet Oncology*). Hence, it rather appears as if cell death is triggered in Ewing sarcoma cells upon knockdown of *PRC1* due to the still relatively high maintained mitotic activity while cytokinesis is largely disturbed, leading to aneuploidy and non-viable karyotypes. However, we agree with this Reviewer that this aspect should have been pointed out more clearly, which is why we have now modified the corresponding Results section:

Page 6, paragraph 1: ‘Notably, although *PRC1* knockdown appeared to strongly reduce tumor growth, the remaining proliferation rate of the tumor cells was still relatively high compared to other malignancies, such as breast cancer (for review see Yerushalmi R *et al.* 2010 *Lancet Oncology*), suggesting that the massive induction of apoptosis rather than the mere blockage of mitotic activity was the main driver of reduced tumor growth.’

Accordingly, while we agree with this Reviewer that the rate of proliferation undoubtedly has effects on the efficacy of chemotherapeutics, there appears to be an additional effect of the *PLK1-PRC1* axis that relies on maintained mitotic activity while cytokinesis is blocked. To elaborate on this aspect, we have now expanded our discussion section in agreement with the proposal of Reviewer #3 (see below). In further support of this notion, it may have escaped to this Reviewer’s notice that we have provided a Supplementary table showing results from a set of cell line derived xenografts from relatively genomically silent cancer types in which we observed that good responses to volasertib (BI6727) were observed exclusively among *PRC1*-high tumors, but that there was no association with the expression levels of the proliferation marker Ki-67 (encoded by *MKI67*). These results are mentioned in the revised Results section:

Page 9, paragraph 1: ‘Together, these findings suggested that genomically silent pediatric cancers, such as EwS, may be very sensitive to PLK1 inhibition in case of high *PRCI* expression. In support of this notion, analysis of matched *in vivo* gene expression and drug-response data from pediatric tumor types (including EwS) (Gorlick, R. *et al.* 2014 *Pediatr. Blood Cancer*) with relatively silent genomes revealed that good responses to BI6727 (Volasertib) were observed exclusively among *PRCI* high expressing xenografts (defined by median expression; $P=0.0325$, Fisher’s exact test) – an effect not observed for *PLK1* and *MKI67* (**Supplementary Table 5**).

Does PLK1 inhibitors show a G2/M blockade with monster cells and aneuploidy in EWS?

Authors: We thank this Reviewer for this interesting question. Indeed, as displayed in the initially submitted Supplementary Fig. 4e (now Supplementary Fig. 4g), we observed that both PLK1 inhibitors led to a significant induction of monster cells and aneuploidy. Strikingly, this effect could be abrogated by genetic KO of the *PRCI*-associated enhancer-like GGAA-mSat. However, we agree with this Reviewer that this fact was not sufficiently explained in the main text. To clarify this important point, we have now modified the Results section as follows:

Page 8, paragraph 1: ‘Treatment of mice xenografted with highly *PRC1* expressing EwS cells led to strong inhibition of tumor growth and even tumor regression after only three cycles of treatment without overt adverse effects, such as weight loss (**Figs. 4e,f, Supplementary Figs. 4e,f**). In addition, both PLK1-inhibitors led to a significant increase in the numbers of aneuploid cells and ‘monster cells’ (**Supplementary Fig. 4g**). Strikingly, this effect could be abrogated by genetic KO of the *PRCI*-associated enhancer-like GGAA-mSat (**Supplementary Fig. 4f,g**).

Given the results reported here, it is disappointing that as reported by Gorlick et al that EWS do not show response to PLK1 inhibitors.

Authors: We fully agree with this Reviewer. However, it should be noted that in the paper by Gorlick *et al.* only 4 different cell line-derived xenografts were examined. These cell lines were likely not pre-selected for *PRC1* expression levels, which could possibly explain the relatively low response rates. Interestingly, the xenograft with the lowest response was derived from the TC71 cell line, which has particularly low *PRCI* expression levels and a rather low average number of consecutive GGAA-repeats at the enhancer-like *PRCI*-associated GGAA-mSat. To accommodate this Reviewer’s comment, we have adapted the discussion section also in agreement with Reviewer #3 (see below):

Page 11, paragraph 1: ‘Notably, our data may also shed new light on why previous preclinical testing of PLK1 inhibition in non-preselected EwS models may have yielded heterogeneous results on its efficacy (Gorlick *et al.* 2014 *Pediatr. Blood Cancer*). In fact, especially the TC71 cell line used in this screen exhibited a relatively low response toward BI6727 and the lowest *PRCI*

expression levels among all four cell lines tested *in vivo* (Gorlick *et al.* 2014 *Pediatr. Blood Cancer*) (Supplementary Table 5). These findings correspond to the rather low *PRC1* expression levels and short *PRC1*-associated GGAA-mSat of the TC71 cell line as demonstrated in the current study (Supplementary Fig. 2b).’

The data generated in this manuscript is sound and does demonstrate that Ewing’s sarcoma is dependent on expression of *PRC1* for survival, and that *PRC1* is a direct target of *EWSR1-FLI1*, and sensitivity to *PLK1* inhibitors is related to *PRC1* expression. However, it is likely that the findings are specific to *EWS* as *PRC1* is an essential gene. In addition, *PLK1* inhibitors have shown disappointing single agent results *in-vivo* models of *EWS*.

Authors: We thank this Reviewer for pointing out the scientific rigor of our results and their mechanistic implications. As explained above, we agree with this Reviewer that a previous report has shown only moderate effects of the *PLK1* inhibitor B6727 (volasertib) as a single agent in non-preselected *in vivo* models of only four Ewing sarcoma cell line-derived xenografts.

To address this aspect, we have already shown in our initially submitted manuscript that two different *PLK1* inhibitors strongly synergize with vincristine (VCR) *in vitro*. To further elaborate this finding, we now provide new data from an *in vivo* drug combination experiment showing that VCR and the *PLK1* inhibitor BI6727 strongly synergize also *in vivo*. In fact, we now demonstrate that addition of VCR allows to strongly reduce (6-fold) the required dose of BI6727 (now 5 mg/kg instead of 30 mg/kg) to still achieve tumor regression. These new data are now shown in the revised Fig. 4 and explained in the Results section:

Page 10, paragraph 1: ‘Similar to the *in vitro* findings, we noted a strong synergistic effect of BI6727 and VCR *in vivo*, even when applying a 6-fold reduced dose of BI6727 (now 5 mg/kg) as inferred from our *in vitro* synergy assays (Supplementary Fig. 5c). In fact, while VCR or BI6727 (at the reduced dose) as single agents only delayed tumor growth, combination of both drugs led to tumor regression in all mice without adverse effects, such as weight loss (Fig. 4h, Supplementary Fig. 5d,e).’

In addition, the corresponding Methods section was modified for this *in vivo* experiment as follows:

Page 33, paragraph 2: ‘For *in vivo* experiments using VCR and/or the *PLK1* inhibitor BI6727 as single agent or in combination, 5×10^6 TC32 EwS cells were subcutaneously injected in mice as described above. When the tumors reached an average volume of $\sim 100 \text{ mm}^3$, mice were randomly distributed in equal groups and henceforth treated with vehicle (0.1N HCl with 0.9% saline), VCR (alone i.p. [1 mg/kg/d] on days 0 and 1 of treatment), BI6727 (Volasertib; alone, i.v. [5 mg/kg] on day 0 of treatment), or VCR (i.p. [1 mg/kg/d] on days 0 and 1 of treatment) plus BI6727 (Volasertib; i.v. [5 mg/kg] on day 0 of treatment) for 4 treatment-cycles. At the experimental endpoint or if humane endpoints as described above were reached before, mice were sacrificed by cervical

dislocation. Then, xenografted tumors were extracted and fixed in 4%-formalin and paraffin-embedded (FFPE) for (immuno)histology. Animal experiments were approved by the government of Upper Bavaria and Northbaden, and conducted in accordance with ARRIVE guidelines, recommendations of the European Community (86/609/EEC), and UKCCCR (guidelines for the welfare and use of animals in cancer research).’

Reviewer #2 remarks to the authors

In this manuscript, (Therapeutic targeting of cytokinesis triggers mitotic catastrophe in genomically silent childhood cancer) Li et al. used an unbiased approach to identify cytokinesis regulatory proteins that promote Ewing Sarcoma malignancy and identified PRC1 as a potential candidate in EwS. They provide compelling evidence for EWSR1-FLI1 targeting a GGAA-mSat for PRC1 expression and establish functional relevance for this mechanism in EwS proliferation and sphere formation. Knockdown of PRC1 induced apoptosis and inhibited cell proliferation, colony formation and tumor growth. Further, PRC1 knockdown induced chromosomal aberrations (aneuploidy) and some cells appeared as “monster cells”. Inhibition of the upstream (negative) PRC1 regulator Plk1 induced antineoplastic effects. Importantly, these effects were diminished after downregulation of PRC1 or knockout of GGAA-mSAT, indicating Plk1 inhibitors exert these effects through PRC1. Consistently, in flank xenograft experiments, both Plk1 inhibitors resulted in inhibition of tumor growth and knockout of the GGAA-mSat partially reversed these effects. In cells, vincristine (but not doxorubicin) acted synergistically with Plk1 inhibitors and synergy was lost after PRC1 knockdown.

In summary, this is a large body of work using thorough and elegant techniques that strongly suggest that EwS with elevated PRC1 may be sensitive to Plk1 inhibition, in particular when combined with vincristine. The results are quite dramatic, and the data are compelling and strongly support the main conclusions. The data are corroborated by extensive supplementary material, robust quantification and solid statistics. A major strength of this study is the fact that in vitro findings are corroborated by extensive flank tumor analysis (mainly IHC), underscoring the translational relevance of the findings reported in this paper.

Authors: We thank this Reviewer for pointing out the scientific rigor, comprehensiveness, novelty, importance and translational relevance of our findings.

However, some conclusions about the mechanism/phenotypes are a bit overstated.

Major comments:

Title: the title is catchy, but this study does not provide thorough analysis of cytokinesis. Also, the term “mitotic catastrophe” is only referred to once in the manuscript (Fig. 4k, schematic) but not investigated experimentally. Therefore, using both terms (cytokinesis and mitotic catastrophe) in the title is not appropriate. Additionally, while the exact mechanism of Plk1 inhibition on PRC1

function is not investigated, the authors clearly establish the functional relevance for the Plk1-PRC1 axis, and this should be indicated in the title.

Authors: We thank this Reviewer for his/her suggestion. Although we further corroborated our investigations on cytokinesis and mitotic defects along this Reviewer's recommendations (see our responses below and new data in the revised version of our manuscript), we have modified the title as suggested, which now reads as follows:

'Therapeutic targeting of the PLK1-PRC1-axis triggers cell death in genomically silent childhood cancer'

According to recent evidence, Plk1 is the major (negative) regulator of PRC1 (Hu C-K. et al. Mol Biol Cell (2012) Jul;23(14):2702). Do the authors have any evidence that expression of an unphosphorylatable PRC1 construct exerts antineoplastic effects similar to Plk1 inhibition? The authors should at least address the potential mechanism of Plk1 inhibition on PRC1 biological activity and on cytokinesis (which is mentioned in the title but not addressed in the manuscript) in the discussion.

Authors: We fully agree with this Reviewer and have now discussed the paper of Hu et al. and the potential mechanism of PLK1 inhibition on PRC1 biological activity in our Discussion section (see also response to a comment of Reviewer #3 below).

Page 11, paragraph 1: 'Notably, our data may also shed new light on why previous preclinical testing of PLK1 inhibition in non-preselected EwS models may have yielded heterogeneous results on its efficacy (Gorlick *et al.* 2014 *Pediatr. Blood Cancer*). In fact, especially the TC71 cell line used in this screen exhibited a relatively low response toward BI6727 and the lowest *PRC1* expression levels among all four cell lines tested *in vivo* (Gorlick *et al.* 2014 *Pediatr. Blood Cancer*) (**Supplementary Table 5**). These findings correspond to the rather low *PRC1* expression levels and short *PRC1*-associated GGAA-mSat of the TC71 cell line as demonstrated in the current study (**Supplementary Fig. 2b**). Yet, it should be noted that the interaction of PLK1 and PRC1 is complex: While PRC1 phosphorylation by PLK1 is required for formation of the PRC1-PLK1 protein complex and its translocation to the spindle midzone, it has been reported that PLK1 can also negatively regulate PRC1 to prevent premature midzone formation before cytokinesis (Hu CK *et al.* 2012 *Mol Biol Cell*). In turn, microtubules can stimulate PRC1 phosphorylation by PLK1, creating a potential negative feedback loop controlling PRC1 activity (Hu CK *et al.* 2012 *Mol Biol Cell*). These facts imply that *PRC1* highly expressing cells may have adapted by increasing PLK1 to dampen the effects of high PRC1, which would be in agreement of our finding that *PRC1* and *PLK1* are significantly co-expressed in patient EwS tumors (**Supplementary Fig. 4a**). Accordingly, it is tempting to speculate that when PLK1 is inhibited, the PRC1 function may become deregulated and toxic to the cell. Although this is subject to future research, it is conceivable that the PRC1-related mechanism identified in our EwS model may be translatable to

other cancers for which immunohistochemical detection of high PRC1 levels could serve as a broadly available, and inexpensive predictive biomarker.’

Fig. 1B,C: The authors show PRC1 prognostic significance but later target Plk1 as the major negative upstream regulator of PRC1. Therefore, what is the prognostic value of Plk1 expression? As Supp Fig. 4A shows correlation of PRC1 and Plk1 in EwS it would be interesting to determine the multi-gene prognostic index (survival) to see whether combined expression of PRC1 and Plk1 is associated with worse prognosis.

Authors: We thank this Reviewer for this interesting question. To address this aspect, we have reassessed our mRNA cohort for the prognostic relevance of *PLK1*. As shown in the new supplementary Fig. 4b, also high *PLK1* mRNA expression showed a statistically significant association with worse overall survival of Ewing sarcoma patients. Accordingly, we have calculated a combined prognostic index of *PRC1* and *PLK1* mRNA expression by using the Cox regression coefficient for these genes in this cohort, which showed a poor overall survival in EwS patients with high co-expression of *PRC1* and *PLK1*. These new data are now displayed in Supplementary Fig. 4c and described in the Results section:

Page 8, paragraph 1: ‘In accordance, both genes were highly significantly co-expressed in patient EwS tumors ($n=196$, $r_{Pearson}=0.58$, $P=2.2^{-16}$) (**Supplementary Fig. 4a**), and combining both markers yielded a highly significant association with worse overall survival ($P=0.0013$) (**Supplementary Fig. 4b,c**).

In addition, the corresponding Methods section was modified as follows:

Page 38, paragraph 1: ‘To develop a prognostic index (PI), EwS patients were first stratified in two groups (i.e., *PRC1*-high and -low, and *PLK1*-high and -low expression) using the median cut-off of the expression level for each gene. The PI of every patient was calculated as the sum of each gene score, which was calculated by multiplying the expression level of a gene by its corresponding coefficient (i.e., $PI = \sum \text{Cox coefficient of gene } G_i \times \text{expression level of gene } G_i$). Subsequently, patients were stratified into three PI groups (i.e., low, moderate, and high) by thirds of the calculated PI as cut-offs. Kaplan-Meier analysis was performed and the difference in survival time between the lowest and highest third was assessed using a Mantel-Haenszel test.’

Fig. 2B: upper panels: It is odd that the horizontal bars are all the same size around 9 Immune Reactive Score. According to the individual dots, at least the median of the third bar should be much higher than 9. Also, in the lower panels, the IHC images are of poor resolution (this is true for most IHC images throughout the manuscript).

Authors: We thank this Reviewer for this comment. We have carefully assessed the data, but found that the median is correctly plotted in the graph. Yet, we agree with this Reviewer that the IHC images of Fig. 2b were of too low magnification. Hence, we have provided higher magnification IHC images for this figure, and also display these IHC images at much higher resolution. As for the other IHC images, we apologize for the poor resolution that may have been arisen from compressing the PDF files during the submission process. We now provide an uncompressed PDF file and believe that the IHC images are now of sufficient resolution when zooming in.

Fig. 3: The notion of “disruption of the delicate balance between mitosis and cytokinesis” is not sufficiently corroborated. I agree the authors show chromosomal aberrations (e.g., aneuploidy), large “monster” cells and increased p-gH2AX. But this is not enough evidence to conclude these phenotypes are the result of an imbalance between mitosis and cytokinesis. Monitoring bipolar spindle formation and contractile ring formation together with chromosome segregation over time would be required in order to make such statements.

Authors: We thank this Reviewer for this important remark. We fully agree with this suggestion and have carried out the recommended experiments. In fact, we performed time-lapse live-cell imaging of TC32 and RDES EwS cells with/without knockdown of *PRCI* and monitored chromosome segregation over time. As shown in the new Fig. 3h, knockdown of *PRCI* was associated in both cell lines with a significantly higher number of EwS cells exhibiting mitotic activity without subsequent cytokinesis leading to chromosome missegregation. These new results are now explained in the Results section:

Page 7, paragraph 1: ‘Moreover, time-lapse live-cell imaging of TC32 and RDES EwS cells with/without *PRCI* knockdown demonstrated that *PRCI* silencing was associated with a significantly higher number of cells exhibiting mitotic activity without subsequent cytokinesis resulting in chromosome missegregation in both cell lines ($P=0.0022$) (Fig. 3h, Supplementary Fig. 3g).’

In addition, we modified the Methods section for these new experiments as follows:

Page 31, paragraph 2: ‘TC32 and RDES EwS cells harboring a Dox-inducible shRNA against *PRCI* were seeded on coated Ibidi μ -slide 8 well (Ibidi GmbH, Germany) (chamber) slides at 300 μ l/well (6×10^4 cells) and incubated at 37°C for 48h with/without Dox (1 μ g/ml) to allow for *PRCI* knockdown. On the day of imaging, cells were washed 3 \times with prewarmed FluoroBrite™ DMEM (A1896701, Life Technologies) and then stained for 30 min with 1X CellMask™ Deep Red Acting Tracking Stains (A57245, Life Technologies) in 250 μ l Live Cell Imaging Solution (A14291DJ, Invitrogen). The staining solution was then carefully removed and the cells were subsequently counterstained with Hoechst 33342 Ready Flow Reagent (RF001, Invitrogen) for 15 min in the dark at 37°C. After exposure, cells were washed 3 \times with Live Cell Imaging Solution at 37°C and

then imaged and analyzed in 300 μ l prewarmed FluoroBrite™ DMEM (A1896701, Life Technologies) supplemented with 10% FCS, 4 mM GlutaMax (Invitrogen), and 20 mM HEPES (Invitrogen). The environment throughout imaging was controlled at 37°C, 5% CO₂, and 90% humidity. Time-lapse images were acquired with an inverted Zeiss Axio Observer Z1 widefield microscope equipped with a piezoelectric focus and an AxioCam MRm gray scale CCD camera, and controlled by ZEN pro software (Zeiss). Brightfield, Cy5 (AHF F36-523) and DAPI (Zeiss Filter Set 49, 488049-9901-000) were captured using a 40 \times oil, 1.4 numerical aperture (NA) objective combined with 2 \times 2 binning mode. Fluorescence and bright-field images were acquired as Z-stacks (10 planes, 2 μ m interval) in 2 \times 2 binning mode at 15–20 random positions every 30 min for 18h per condition. The resulting images were processed, analyzed and colored using ZEN pro (Zeiss) software. When necessary, signals were enhanced for optimal contrast using Adobe Photoshop 2020. The percentage of mitotic cells that exited mitosis as a single cell were reported as those that fail cytokinesis.’

Supp Fig. 3f: The notion of higher numbers of tetraploid cells through blockage of proper G2/M transition (page 6) is not supported by the data shown. I see no evidence of tetraploid cells in Supp Fig. 3. Also, any statement about G2/M transition needs to be corroborated by analyzing cycling cells, which can be done by releasing cells from synchronization (hydroxyurea or double thymidine) and monitoring cell cycle by Flow and/or expression of cell cycle markers (e.g., cyclins) over time. As I cannot find any synchronization of cells in the figure legend or methods regarding the experiment in Supp Fig. 3, the results in Supp Fig. 3f are only a snapshot in time and no conclusions about cell cycle transitions can be drawn. In addition, the siCtrl is missing, making it difficult to draw any conclusions.

Authors: We fully agree with this Reviewer that synchronization experiments are more appropriate to corroborate statements on the G2/M transition. Accordingly, we have – as suggested by this Reviewer – replaced our unsynchronized cell cycle experiments by those with synchronized cells using a double thymidine block/release. In these new experiments, which now also include control cells (shCtrl), synchronized cells were fixed at different time points (24h, 48h, 72h) and their DNA content was analyzed by PI-staining and flow cytometry. These experiments revealed that the shRNA-mediated knockdown of *PRCI* is associated in three Ewing sarcoma cell lines with a statistically significant increase of the fraction of cells being in G2/M phase over time as compared to shCtrl cells. Also, we now specify in the corresponding Figure Legend that the reported *P*-values in each condition refer to statistical differences in the G2/M phases (see revised Supplementary Figs. 3f). Lastly, we also agree with this Reviewer that the mere demonstration of a blockage in G2/M transition possibly does not allow to conclude that all cells will maintain a permanent tetraploidy. Thus, we have modified the Results section as follows:

Page 7, paragraph 1: ‘Indeed, cell cycle analysis of synchronized EwS cells showed that *PRCI* silencing led to a higher fraction of cells in G2/M phase over time, which is indicative of a delayed

transition through G2/M-phase that may contribute to the generation of tetraploid cells (**Supplementary Fig. 3f**).’

The modified the Method section as follows:

Page 28, paragraph 2: ‘For analysis of cell cycle, RDES, SK-N-MC, and TC32 cells harboring a Dox-inducible shRNA against *PRCI* and respective controls were synchronized by a double thymidine (T1850, Sigma-Aldrich) block/release as previously described with slight modifications (Chen, G. & Deng, X. 2018 *Bio-Protoc.*). Briefly, cells were blocked in G1/S with 1 mM thymidine for 18h at 37°C, then released into S phase by washing 3× with pre-warmed serum free media. Fresh complete medium was added to the cells and incubated for 10h at 37°C. Second round of thymidine to a final concentration of 1 mM was added and cells were cultured for another 18h at 37°C. Cells are now in G1/S boundary. Cells were in G1/S boundary by then. After washing 3× with pre-warmed serum free media, cells were seeded at 5×10⁵ cells per T25 flask in the fresh medium with/without addition of Dox (1 µg/ml). Dox was renewed 48h after seeding. Cells were fixed with ice-cold 70% ethanol at each time point post releasing (24h, 48h, and 72h), treated with 100 µg/ml RNase (ThermoFisher) and stained with 50 µg/ml propidium iodide (PI) (Sigma-Aldrich).’

Fig. 4g and related figures: what was the experimental readout in these experiments to assess synergy (Bliss score)? I assume these were *in vitro* experiments. Please clarify in figure legends.

Authors: We apologize for this confusion. Indeed, these were *in vitro* experiments, which has now been clearly stated in the Figure Legend.

In addition, we now provide new data from an *in vivo* drug combination experiment, which showed that vincristine (VCR) and the PLK1 inhibitor BI6727 (volasertib) strongly synergize *in vivo* (see also our response to a comment of Reviewer #1 above). In fact, we now demonstrate that addition of VCR allows to strongly reduce (6-fold) the required dose of BI6727 (now 5 mg/kg instead of 30 mg/kg) to still achieve tumor regression. These new data are now shown in the revised Fig. 4 and explained in the Results section:

Page 10, paragraph 1: ‘Similar to the *in vitro* findings, we noted a strong synergistic effect of BI6727 and VCR *in vivo*, even when applying a 6-fold reduced dose of BI6727 (now 5 mg/kg) as inferred from our *in vitro* synergy assays (**Supplementary Fig. 5c**). In fact, while VCR or BI6727 (at the reduced dose) as single agents only delayed tumor growth, combination of both drugs led to tumor regression in all mice without adverse effects, such as weight loss (**Fig. 4h, Supplementary Fig. 5d,e**).’

Accordingly, the corresponding Methods section was modified for this *in vivo* experiment:

Page 33, paragraph 2: ‘For *in vivo* experiments using VCR and/or the PLK1 inhibitor BI6727 as single agent or in combination, 5×10^6 TC32 EwS cells were subcutaneously injected in mice as described above. When the tumors reached an average volume of $\sim 100 \text{ mm}^3$, mice were randomly distributed in equal groups and henceforth treated with vehicle (0.1N HCl with 0.9% saline), VCR (alone i.p. [1 mg/kg/d] on days 0 and 1 of treatment), BI6727 (Volasertib; alone, i.v. [5 mg/kg] on day 0 of treatment), or VCR (i.p. [1 mg/kg/d] on days 0 and 1 of treatment) plus BI6727 (Volasertib; i.v. [5 mg/kg] on day 0 of treatment) for 4 treatment-cycles. At the experimental endpoint or if humane endpoints as described above were reached before, mice were sacrificed by cervical dislocation. Then, xenografted tumors were extracted and fixed in 4%-formalin and paraffin-embedded (FFPE) for (immuno)histology. Animal experiments were approved by the government of Upper Bavaria and Northbaden, and conducted in accordance with ARRIVE guidelines, recommendations of the European Community (86/609/EEC), and UKCCCR (guidelines for the welfare and use of animals in cancer research).’

It should be noted that for technical reasons the detection system of IHC staining used in the VCR-PLK1 combination *in vivo* experiment was changed, now employing a red chromogen instead of a brown chromogen. The Methods section was adapted accordingly.

In general, most IHC images are of very poor resolution when zoomed in. Please provide images of better quality.

Authors: As stated above, we apologize for the poor resolution that may have been arisen from compressing the PDF file for submission. Whenever possible, we tried to enhance the magnification of the IHC images, such as in the revised Fig. 2b, and now provide an uncompressed PDF file. Also, we now provide the Supplementary Figures as a separate file to enable larger file sizes of the individual files.

I am not familiar with the term “massive genomic chaos” (page 8). Authors need to provide a definition for this term and additionally thorough scientific evidence to corroborate this statement.

Authors: We fully agree with this Reviewer that this term was confusing and have adapted and specified this part of the Results section as follows:

Page 9, paragraph 1: ‘In addition, both PLK1-inhibitors led to a significant increase in the number of aneuploid cells and ‘monster cells’ (**Supplementary Fig. 4g**). Strikingly, this effect could be abrogated by genetic KO of the *PRCI*-associated enhancer-like GGAA-mSat (**Supplementary Fig. 4f,g**).

Page 6, line 2: the word “proven” should be replaced (e.g., monitored, confirmed).

Authors: We agree with this Reviewer and have adapted the text as follows:

Page 6, paragraph 1: ‘Knockdown efficacy was confirmed by qRT-PCR and western blotting (Supplementary Figs. 3a,b).’

Minor comments:

Immune Reactive Score (IRS) is an established technique for estrogen and progesterone receptors in breast cancer, but it is not widely accepted for Ewing Sarcoma (only one Oncotarget paper from the same group). I’m not a pathologist expert but I recommend a board-certified pathologist should elaborate why this technique is superior to other pathological methodologies and that its use is justified here for Ewing Sarcoma.

Authors: We politely disagree with this Reviewer. As stated in the Methods section, the IRS as used here was ‘in analogy’ to the classical IRS as employed for hormone receptor scoring, but it is not identical, which is explained in detail in the Methods section. Indeed, while the classical IRS has only a 5-tier grading with uneven stratification of the percent of positive cells, our modified IRS has an evenly stratified and thus more fine-grained grading of the percentage of positive cells. This modified IRS used in the current paper has been carefully adapted to the requirements of Ewing sarcoma by several board-certified reference pathologists co-authoring this paper and been used in many more publications as follows:

- Baldauf MC et al 2018 Oncotarget
- Baldauf MC et al 2018 OncoImmunology
- Dallmayer M et al. 2019 Cell Death Dis.
- Orth MF et al 2020 Cancers
- Marchetto A et al. 2020 Nat Commun

Also, it should be noted that the senior PI and corresponding author of this paper is an advanced pathology resident and that two additional board-certified pathologists (Prof. Thomas Kirchner, director of the Institute of Pathology of the LMU Munich, Germany; Prof. Wolfgang Hartmann, vice-director of the Gerhard-Domagk-Institute for Pathology of the University of Münster, Germany) have provided histological guidance (as stated in the author contribution section).

Beyond this, the quoted Oncotarget paper by Baldauf *et al.* 2018 was fully confirmed in the subsequent validation study by Orth *et al.* in 2020 using the exact same modified IRS scoring method. Both papers concerned the diagnostic utility of IHC markers in Ewing sarcoma.

Hence, we do believe that this scoring method is fully adequate to assess the PRC1 expression levels in Ewing sarcoma tumors. However, to address this Reviewers comment, we have modified the Methods section accordingly and added more references:

Page 35, paragraph 2: ‘Evaluation of PRC1 immunoreactivity was carried out in analogy to scoring of hormone receptor Immune Reactive Score (IRS) ranging from 0–12. This modified IRS scoring scheme has been adapted to EwS and been described and validated previously (Baldauf MC *et al.* 2018 *Oncotarget*, Baldauf MC *et al.* 2018 *OncoImmunology*, Dallmayer M *et al.* 2019 *Cell Death Dis.*, Orth MF *et al.* 2020 *Cancers*, Marchetto A *et al.* 2020 *Nat Commun*).’

For sphere formation: spheroid area was calculated using $A=\pi A \sim d^2/4$ but some spheres are not perfectly round. How was this taken into account? Was diameter taken from two different (orthogonal) measurements?

Authors: We apologize for this confusion. The initially provided formula was not entirely accurate. Indeed, as indicated by this Reviewer, the diameters were taken from two orthogonal measurements to account also for elliptical spheres. To point this out more clearly, the correct formula is now provided in the revised Methods section:

Page 27, paragraph 2: ‘At day 14, wells were photographed, and sphere numbers as well as the spheroid areas were analyzed using ImageJ using the formula: $\text{Area}=\pi \times A \times B/4$ (with A and B referring to orthogonal diameters)’

Page 8: “veil injection” should be “vein injection”.

Authors: We apologize for this typo that has now been corrected.

Reviewer #3 remarks to the authors:

This reports an intriguing and potentially important finding in the pathobiology of Ewing Sarcoma and the role of PRC1 as a novel direct target of EWSR1-FLI1 transactivation. The approach is logical and systematic, the experiments are well designed, and the data compelling.

Authors: We thank this Reviewer for pointing out the novelty and importance of our findings as well as the scientific rigor of our results.

However, some broader issues or questions remain to be addressed:

1. Other alterations are well established as prognostic in Ewing Sarcoma, including TP53 mutations and, especially, STAG2 loss. How does the PRC1 high subset overlap with the STAG2 loss subset?

Authors: We thank this Reviewer for this important question. Part of our transcriptome cohort that we used as discovery cohort for the evaluation of the prognostic significance of *PRCI* was part of the ICGC study on Ewing sarcoma (Tirode *et al.* 2014 *Cancer Discov*) mentioned by this Reviewer below. While our manuscript was under peer-review new data were published that enabled us to analyze the overlap of *STAG2*- and *TP53*-mutated cases in the *PRCI* high expressing subset. This dataset comprised 57 ICGC cases for which matched RNA-sequencing and mutation data were available (Petrizzelli *et al.* 2021 *Methods Mol Biol*). Stratifying this cohort like our discovery cohort in thirds regarding *PRCI* expression, we noted a trend for an increasing overlap with *STAG2*- or *TP53*-mutated case. However, in both instances this trend did not reach statistical significance. These new findings are now presented in the new Supplementary Table 6 and explained in the results section as follows:

Page 10, paragraph 1: ‘Along the same lines, there was no statistically significant overlap between *STAG2*- or *TP53*-mutated and *PRCI* highly expressing patient EwS tumors (**Supplementary Table 6**).’

The Methods section was adapted accordingly:

Page 38, paragraph 1: ‘Potential associations of the *STAG2* and/or *TP53* mutation status with *PRCI* expression levels was assessed by a two-sided Fisher’s exact test in 57 ICGC EwS cases for which the mutation status and transcriptome profiles were available (Petrizzelli *et al.* 2021 *Methods Mol Biol*).’

Based on Table 1 in PMID 25223734, all the Ewing lines used are *TP53*-mutated/*STAG2*-wt, except SK-N-MC which is a double mutant. This is potentially significant and possibly confounding given that *PLK1* also phosphorylates *STAG1/2* to enable dissociation of the cohesin complex (PMID 31516082 and PMID 15737063).

Authors: We thank this Reviewer for this important remark. However, we politely disagree with this Reviewer. We have used for the majority of our experiments the following four cell lines: A673, RDES, SK-N-MC, and TC32. The cell line TC32 has not been analyzed in the mentioned paper by Tirode *et al.* 2014 *Cancer Discov* (PMID 25223734), but is a known *STAG2*-mutated, *TP53*-wt EwS cell line (Brohl *et al.* 2014 *PLoS Genetics*, PMID 25010205). As shown in the initially submitted Supplementary Fig. 5 (now Supplementary Fig. 6), we observed almost identical effects of two different *PLK1* inhibitors depending on the *PRCI* knockdown status in TC32 (*STAG2*-mut, *TP53*-wt) and RDES (*STAG2*-wt, *TP53*-mut). Hence, we have no evidence from our results that our phenotypes may have been confounded by the *STAG2* and/or *TP53* mutation status. However, to accommodate this Reviewers concern, we have mentioned this aspect and quoted the suggested references in the revised Results section:

Page 10, paragraph 1: ‘It should be noted that almost identical effects on the PRC1-dependent PLK1 efficacy were observed in TC32 and RDES cells (**Fig.4g, Supplementary Fig. 5a**), which differ in their *STAG2* and *TP53* mutation status (TC32: *STAG2*-mut/*TP53*-wt; RDES: *STAG2*-wt/*TP53*-mut) (Tirode *et al.* 2014 *Cancer Discov*; Brohl *et al.* 2014 *PLoS Genetics*), suggesting that both mutations have likely no impact on the efficacy of PLK1 inhibition in EwS cells, although it has been reported that PLK1 can phosphorylate *STAG1/2* (Piché *et al.* 2019 *Cell Cycle Georget.*, Hauf *et al.* 2005, *PLoS Biol.*). Along the same lines, there was no statistically significant overlap between *STAG2*- or *TP53*-mutation and high *PRC1* expression in patient EwS tumors (**Supplementary Table 6**).’

2. Although PLK1 does interact with PRC1, that interaction is not so simple. According to PMID 22621898, PLK1 negatively regulates PRC1 through phosphorylation. In turn, microtubules can stimulate PRC1 phosphorylation by PLK1, creating a potential negative feedback loop controlling PRC1 activity. The authors should comment on this aspect. For instance, perhaps PRC1-high cells have adapted by increasing PLK1 to dampen the effects of high PRC1; when PLK1 is inhibited, PRC1 function becomes totally deregulated and toxic to the cell.

Authors: We thank this Reviewer for bringing these important aspects to our attention. We fully agree with this Reviewer and have adapted the Discussion section accordingly and quoted the mentioned reference (also in agreement with a comment of Reviewer #2, see above). In fact, the molecular mechanism and adaptation process proposed by this Reviewer is in line with our initially presented data of patient Ewing sarcoma tumors in which we observed a highly significant co-expression of *PRC1* and *PLK1* (Supplementary Fig. 4a):

Page 11, paragraph 1: ‘Notably, our data may also shed new light on why previous preclinical testing of PLK1 inhibition in non-preselected EwS models may have yielded heterogeneous results on its efficacy (Gorlick *et al.* 2014 *Pediatr. Blood Cancer*). In fact, especially the TC71 cell line used in this screen exhibited a relatively low response toward BI6727 and the lowest *PRC1* expression levels among all four cell lines tested *in vivo* (Gorlick *et al.* 2014 *Pediatr. Blood Cancer*) (**Supplementary Table 5**). These findings correspond to the rather low *PRC1* expression levels and short *PRC1*-associated GGAA-mSat of the TC71 cell line as demonstrated in the current study (**Supplementary Fig. 2b**). Yet, it should be noted that the interaction of PLK1 and PRC1 is complex: While PRC1 phosphorylation by PLK1 is required for formation of the PRC1-PLK1 protein complex and its translocation to the spindle midzone, it has been reported that PLK1 can also negatively regulate PRC1 to prevent premature midzone formation before cytokinesis (Hu CK *et al.* 2012 *Mol Biol Cell*). In turn, microtubules can stimulate PRC1 phosphorylation by PLK1, creating a potential negative feedback loop controlling PRC1 activity (Hu CK *et al.* 2012 *Mol Biol Cell*). These facts imply that *PRC1* highly expressing cells may have adapted by increasing PLK1 to dampen the effects of high PRC1, which would be in agreement of our finding that *PRC1* and *PLK1* are significantly co-expressed in patient EwS tumors (**Supplementary Fig. 4a**). Accordingly, it is tempting to speculate that when PLK1 is inhibited, the PRC1 function may

become deregulated and toxic to the cell. Although this is subject to future research, it is conceivable that the PRC1-related mechanism identified in our EwS model may be translatable to other cancers for which immunohistochemical detection of high PRC1 levels could serve as a broadly available, and inexpensive predictive biomarker.’

3. The authors should mention and comment on the poor response of TC71 *in vitro* and *in vivo* to the same PLK Inhibitor (BI6727) when tested by the Pediatric Preclinical Testing Program (PMID 23956067 and their ref. 22). It is not sufficient to just say "previous preclinical testing of PLK1 inhibition in non-preselected EwS models may have yielded controversial results on its efficacy" when the PRC1 status of TC71 is known.

Authors: We fully agree with this Reviewer. Indeed, Gorlick *et al.* (2014 *Pediatr. Blood Cancer*) compared the response of TC71 on BI6727 to three additional Ewing sarcoma cell lines *in vitro* and *in vivo*. However, since in each setting two out of the three additionally tested Ewing sarcoma cell lines were different (*in vitro*: CHLA-9, CHLA-10, CHLA-258, *in vivo*: SK-NEP-1, EW5, CHLA-258), the *in vitro* and *in vivo* data from this study are not fully comparable. Given this limitation, we prefer at the current stage to discuss only the preclinical *in vivo* data of Gorlick *et al.* because these data appear to us more relevant for our translational study. Hence, following this Reviewer’s suggestion, we have now added more details to the Discussion section and modified the above-mentioned sentence as described below:

Page 11, paragraph 1: ‘Notably, our data may also shed new light on why previous preclinical testing of PLK1 inhibition in non-preselected EwS models may have yielded heterogeneous results on its efficacy (Gorlick *et al.* 2014 *Pediatr. Blood Cancer*). In fact, especially the TC71 cell line used in this screen exhibited a relatively low response toward BI6727 and the lowest *PRC1* expression levels among all four cell lines tested *in vivo* (Gorlick *et al.* 2014 *Pediatr. Blood Cancer*) (**Supplementary Table 5**). These findings correspond to the rather low *PRC1* expression levels and short *PRC1*-associated GGAA-mSat of the TC71 cell line as demonstrated in the current study (**Supplementary Fig. 2b**).’

Reviewers' Comments:

Reviewer #1:

Remarks to the Author:

The revised manuscript remains somewhat problematic:

1.It is not surprising that PRC1 is overexpressed in EWS compared to normal terminally divided organ tissue and all cancers will show overexpression compared with normal tissues.

2.PRC1 is a common essential gene (see DepMap data), and all cell lines will likely show sensitivity to its knock out.

3.PLK1 is also a common essential gene and has not shown efficacy in Ewing's sarcoma. The authors would need to do a larger PDX study to validate this given the negative result in the literature.

4.The authors have not convincingly shown the direct link between PLK1 inhibition and PRC1 expression. The continued emphasis that downregulation of PRC1 diminishes the sensitivity of EWS cells toward both PLK1 inhibitors as evidence that EWS is sensitive to PLK1 inhibitors via suppression of PRC1 remains misleading given that EWS show marked apoptosis to suppression of PRC1. The statement " findings suggested that EwS cells with high PRC1 expression are very sensitive to PLK1 inhibition, and that this sensitivity can be almost abolished by suppression of PRC1" is misleading again given the evidence that the authors give that EWS show marked apoptosis to suppression of PRC1.

5.The statement "in vivo gene expression and drug-response data from pediatric tumor types (including EwS) with relatively silent genomes revealed that good responses to BI6727 (Volasertib) were observed exclusively among PRC1 high expressing xenografts (defined by median expression; $P=0.0325$, Fisher's exact test)" is misleading as none of the good responders were EWS tumors. Also, the sample number is low and the significance would disappear if corrected for multiple comparisons.

6.The statement "the rather low PRC1 expression levels and short PRC1-associated GGAA-mSat of the TC71 cell line as demonstrated in the current study" has not been shown, as by mRNA level is very highly expressed in TC71. The authors would need to do a Western of a panel of cell lines to confirm this as I suspect that TC71 will have high expression.

7.The authors could look at much of the drug data e.g. NCI60, Broad data to confirm that the sensitivity to PLK1 is correlated with expression or PRC1.

Reviewer #2:

Remarks to the Author:

The authors have addressed all my comments and concerns satisfactorily.

Reviewer #3:

None

Step-by-step responses to the Reviewers' comments:

Reviewer #2 (Remarks to the Author):

The authors have addressed all my comments and concerns satisfactorily.

Authors: We thank this Reviewer, whose extensive and very constructive questions/comments have very significantly helped us to improve our manuscript. We appreciate that this Reviewer is now pleased with the comprehensive revisions that have been made.

Reviewer #1 (Remarks to the Author):

Authors: We thank this Reviewer for his/her time spent with our manuscript and for the additional comments.

The revised manuscript remains somewhat problematic:

1. It is not surprising that *PRC1* is overexpressed in EWS compared to normal terminally divided organ tissue and all cancers will show overexpression compared with normal tissues.

Authors: We thank this Reviewer for sharing his/her opinion. To explore this possibility, we carried out additional analyses and compared the median *PRC1* expression levels of 40 additional tumor types to that of normal tissues in our well-curated gene expression database (Baldauf *et al.* 2018 OncoImmunology), which we have also used for our initial analyses mentioned by this Reviewer (see **Fig. 1a**). This analysis revealed that the very strong statement of this Reviewer may constitute an overstatement since, although many tumor types including Ewing sarcoma (EwS) show a very significant overexpression of *PRC1* compared to normal tissues, around 20% of analyzed cancer types (including prostate carcinoma and brain cancers) do not (see Figure below).

Despite this finding may be of potential interest for a limited subgroup of readers, we believe that it would contribute little additional information to the main aspects of our manuscript, which is why we would prefer to not further elaborate on this topic at the current stage.

Figure legend: Displayed are *Bonferroni* adjusted *P*-values of *PRC1* mRNA expression levels in 41 cancer entities compared to normal tissues (comprising 71 normal tissue types) from our well-curated gene expression database (Baldauf *et al.* 2018 OncoImmunology). Cancer entities with non-significant overexpression of *PRC1* compared to normal tissues have been marked in green color. EwS has been indicated by an arrow and highlighted in red color.

2. *PRC1* is a common essential gene (see DepMap data), and all cell lines will likely show sensitivity to its knock out.

Authors: We thank this Reviewer for sharing his/her opinion. Indeed, *PRC1* is a common essential gene when considering only the DepMap knockout data. Yet, we believe that such comparison to our data is not fully appropriate since we have not carried out a knockout of *PRC1* in EwS cells.

Indeed, instead of carrying out knockout experiments that would deplete cells in an unphysiological way completely and permanently from *PRC1* protein, we strived for a more physiological approach to mimic the range of *PRC1* protein expression observed in EwS primary tumors (compare **Fig. 1c**). To that end, we did not knockout *PRC1* directly, but rather silence its expression by either targeting *PRC1* mRNA via shRNAs or by modulating the genetic architecture of its associated enhancer-like GGAA-microsatellite. For example, as shown in the initial **Figs. 2e and 4f** around 30% of EwS cells still display *PRC1* expression despite knocking out or epigenetically silencing its enhancer (i.e., CRISPR interference). Interestingly, when comparing our more physiological approach with additional DepMap data

from combined knockdown screens, *PRCI* is **not** classified as a common essential gene in any cancer type including bone sarcomas such as EwS (see Figure derived from the DepMap data portal below).

Yet, we feel that this aspect adds little information to the main messages of our manuscript and that further elaboration on the above-mentioned aspects may rather distract the reader, which is why we would prefer to not further discuss them in our manuscript.

Figure legend: Screenshot from the DepMap data portal concerning *PRCI* (<https://depmap.org/portal/gene/PRCI?tab=overview>) and different screening methods. *PRCI* is classified as a common essential gene when considering the CRISPR knockout screens (Gene Effect score lower than -1), but not in the RNA interference (RNAi) knockdown screens (Gene Effect greater than -1). Data censoring on May 25th 2021.

However, since we believe that it may have escaped to this Reviewer’s notice that we did not carry out a knockout of *PRCI* itself, but rather achieved a **knockdown** by knocking out or epigenetically silencing its enhancer, and to avoid further confusion, we modified the main text of our revised version to further emphasize and clarify this aspect.

Page 5, paragraph 2: “The relationship between *PRCI* and this GGAA-mSat was further probed by CRISPR Cas9-initiated homologous deficiency repair (HDR) DNA editing. To avoid knocking out *PRCI* itself, which may have led to unphysiologically low or absent *PRCI* expression levels, we chose to knockout (KO) its associated enhancer-like GGAA-mSat. Similar to CRISPRi, the genetic (KO) of this GGAA-mSat was accompanied by significantly lower *PRCI* expression levels, proliferation, and sphere-formation in both cell lines (Supplementary Figs. 2d–h).”

3. *PLK1* is also a common essential gene and has not shown efficacy in Ewing's sarcoma. The authors would need to do a larger PDX study to validate this given the negative result in the literature.

Authors: We thank this Reviewer for sharing his/her opinion. Although *PLK1* is classified as a common essential gene in the knockout screen of the DepMap project, we feel that a direct comparison of drug-sensitivity and knockout data is somewhat problematic and perhaps misleading since the responsiveness of a given tumor type toward a give drug relies on multiple factors than the mere target gene expression. For instance, if a given drug is simply not very potent, it is obvious that the gene may incorrectly not be considered as a good drug candidate in screening experiments, although the target per se may be valid. Also, one can only infer from drug screening data on the validity of a gene as a suitable drug target if virtually all applied drugs in a given screen would have similar or even identical pharmacological features (such as, drug stability) and potencies. Since such comparability across drugs is virtually never achieved in any screen, we feel that a direct conclusion from drug screens to knockout screens in which all genes are knocked out equally across cell lines is not fully appropriate.

Given these facts, the above-mentioned inference of this Reviewer from a knockout screen to an unrelated single drug screen only including a single PLK1 inhibitor applied in just a single dosing regimen to only 4 EwS cell lines that were not preselected by any potentially predictive biomarker (see Gorlick *et al.* 2014 Ped Blood Cancer) appears not fully convincing for us.

Instead, we showed with our data from extensive functional experiments that if EwS cells are preselected for *PRC1*-high expression that two independent PLK1 inhibitors can induce full tumor regression *in vivo*. These findings are in line with new data from the DepMap project that are further explained in detail below (see answer to comment #7). Indeed, these new DepMap data showed a significant correlation of *PRC1* expression levels in EwS cell lines with their sensitivity toward the PLK1 inhibitor BI6727 (volasertib), which further confirms one of our main messages of our manuscript that *PRC1* may serve as a predictive biomarker for PLK1 inhibition.

The direct comparison of *PLK1* mRNA expression levels or its knockout with sensitivity toward PLK1 inhibitors, as suggested by this Reviewer, is further problematic because these PLK1 inhibitors do not act at the mRNA level by, e.g., suppressing *PLK1* transcription, but rather inhibiting its kinase function, which cannot be inferred at all from its mRNA expression levels.

In this regard, it may have escaped to this Reviewer's notice that we have already shown in the **initial Supplementary Table 5** that the mRNA levels of *PLK1* did correlate with sensitivity of cell line-derived xenografts toward treatment with BI6727 (Volasertib).

However, to further strengthen this aspect, we analyzed – as suggested by this Reviewer – publicly available gene expression and drug-response data from the DepMap project for EwS cell lines, which showed no correlation of *PLK1* mRNA levels with sensitivity toward Volasertib. Similarly, the expression levels of the proliferation marker *MKI67* did not correlate with sensitivity toward Volasertib in these DepMap data. These new results are now shown in the **new Supplementary Fig. 4I** and **new Supplementary Table 6**, and mentioned in the revised text. Please see also our detailed answer to comment #7 below.

Page 10, paragraph 1: “Together, these findings suggested that genomically silent pediatric cancers, such as EwS, may be very sensitive to PLK1 inhibition in case of high *PRCI* expression. In support of this notion, analysis of matched *in vivo* gene expression and drug-response data from pediatric tumor types (including EwS) (Gorlick *et al.* 2014 Ped Blood Cancer) with relatively silent genomes revealed that good responses to BI6727 (Volasertib) were observed exclusively among *PRCI* high expressing xenografts (defined by median expression; $P=0.0325$, Fisher's exact test) – an effect not observed for *PLK1* and *MKI67* (**Supplementary Table 5**). However, since this dataset contained xenografts from only 4 different EwS cell lines, we extended our analyses using publicly available drug-response and gene expression data from the DepMap project, comprising 11 EwS cell lines (<https://depmap.org>). In this dataset, we observed a relatively strong negative correlation ($r_{Pearson} = -0.54$) of *PRCI* mRNA levels with lower cell viability upon Volasertib treatment ($P=0.04$) (**Supplementary Table 6**). This correlation even remained significant when focusing on the 9 of 11 EwS cell lines that exhibited a confirmed *EWSR1-FLII* fusion ($r_{Pearson} = -0.72$, $P=0.02$) (**Supplementary Fig.4 I**). Similar to our observations made in the *in vivo* dataset (**Supplementary Table 5**) (Gorlick *et al.* 2014 Ped Blood Cancer), such correlations were neither observed for *PLK1* nor *MKI67* regardless of the *EWSR1-FLII* status (*PLK1*: $r_{Pearson} = -0.19/-0.4$, $P=0.29/0.14$; *MKI67*: $r_{Pearson} = 0.04/-0.09$, $P=0.46/0.41$) (**Supplementary Table 6**).”

Although we agree with this Reviewer that validation of our results in a larger PDX study would possibly constitute a potentially interesting addition, we believe that this aspect goes beyond our extensive mechanistic and original discovery study presented here, and that it will be part of future validation studies.

4. The authors have not convincingly shown the direct link between PLK1 inhibition and PRC1 expression.

Authors: We thank this Reviewer for sharing his/her opinion. To address this aspect, we have now added new data from the DepMap project showing a significant correlation between the *PRCI* expression levels and sensitivity toward the PLK1 inhibitor BI6727 (Volasertib) across multiple EwS cell lines. Please see also our replies above and below to questions #3 and #7, respectively.

The continued emphasis that downregulation of PRC1 diminishes the sensitivity of EWS cells toward both PLK1 inhibitors as evidence that EWS is sensitive to PLK1 inhibitors via suppression of PRC1 remains misleading given that EWS show marked apoptosis to suppression of PRC1. The statement “findings suggested that EwS cells with high PRC1 expression are very sensitive to PLK1 inhibition, and that this sensitivity can be almost abolished by suppression of PRC1” is misleading again given the evidence that the authors give that EWS show marked apoptosis to suppression of PRC1.

Authors: We thank this Reviewer for this important remark. Indeed, it is necessary to mention and experimentally prove that the EwS cells were not apoptotic at time of beginning of treatment with PLK1 inhibitors (24–48h after induction of the *PRCI* knockdown or seeding, depending on the assay). To address this aspect, we carried out additional experiments, which showed that at time of beginning of PLK1 inhibition, around ~90–95 % of cells were viable in both the control and treatment groups, and that the rates of apoptotic cells were similarly low (~5–10%) in both groups and statistically not significantly different. These new experiments have been carried out for both the shRNA-mediated *PRCI* knockdown experiments as well as for drug-response assays using EwS cells with a genetic knockout of the *PRCI*-associated GGAA-mSat. These new data rule out the possibility mentioned by this Reviewer that perhaps an already high rate of apoptotic cell death might have biased our results.

In addition, it should be noted that marked apoptosis upon *PRCI* silencing is first observed at much later stages upon *PRCI* silencing (>72h; see initial **Fig. 3c**, and long-term *in vivo* assays shown in **Fig. 3f**), and that this is not an early event being present at the time of start of PLK1 inhibitor treatment. These new data are now explained in the revised text and shown in the new **Supplementary Figs. 4d,e**.

Page 8, paragraph 2: “Importantly, it should be noted that the percentage of viable cells at time of beginning of PLKi-treatment was ~90–95% and that the percentage of apoptotic cells was equal in the control and treatment groups (**Supplementary Figs. 4d,e**).”

The fact that our data was not biased due to unequal rates of cell death across groups is further supported by our method used for the drug-response assays. Indeed, we carried out our drug-response assays with the substance Resazurin, which needs to be critically metabolized by living cells to yield a detectable signal for read-out, and which cannot create a fluorescence emission from dead cells. Moreover, since all drug-response data were normalized to the signal of the corresponding DMSO control of each treatment group, we further can rule out a bias.

This method is comprehensively explained in a recent Methods article from our laboratory with careful and specific adaptations to EwS cells (Musa and Cidre-Aranaz, Drug screening by Resazurin colorimetry in Ewing Sarcoma; *Methods Mol Biol.* 2021;2226:159-166), which was published while our paper was under revision.

To point out these important methodologic facts more clearly and to explain the above-mentioned additional apoptosis assays, we have modified the Methods section and quoted the corresponding reference as follows:

Page 36, paragraph 2: **Drug-response assays and drug combination analysis**

“At the experimental endpoint, cell growth inhibition was assessed using a Resazurin assay (Sigma-Aldrich) with careful adaptations to EwS cells (Musa and Cidre-Aranaz 2021 *Methods Mol Biol*). It should be noted that the dye Resazurin can induce fluorescence emission only in viable cells (Musa and Cidre-Aranaz 2021 *Methods Mol Biol*). The relative IC50 concentrations were calculated using PRISM 8 (GraphPad Software Inc., CA, USA) and normalized to the respective DMSO controls.”

Page 29, paragraph 1: **Cell cycle and apoptosis analysis**

“For time-lapse apoptosis analysis, RDES and TC32 cells harboring a Dox-inducible shRNA against *PRC1* and respective controls were seeded at 8×10^5 cells per 10 cm dish with/without addition of Dox (1 $\mu\text{g/ml}$) and analyzed at different time points after shRNA-mediated knockdown (24h and 48h). The CRISPR Cas9-initiated HDR edited A673 EwS cells and A673 wt cells were seeded at 1×10^6 cells per 10 cm dish and analyzed 24h after seeding. Analysis of apoptosis has been performed at indicated time points by combined Annexin V-FITC/PI

staining (BD Pharmingen FITC Annexin V Apoptosis Detection Kit II). Samples were assayed with BD Accuri C6 Cytometer (BD Biosciences).”

5. The statement “*in vivo* gene expression and drug-response data from pediatric tumor types (including EwS) with relatively silent genomes revealed that good responses to BI6727 (Volasertib) were observed exclusively among PRC1 high expressing xenografts (defined by median expression; $P=0.0325$, Fisher’s exact test)” is misleading as none of the good responders were EWS tumors. Also, the sample number is low and the significance would disappear if corrected for multiple comparisons.

Authors: We thank this Reviewer for sharing his/her opinion. However, we politely disagree with this Reviewer’s comment that our statement would be misleading since all data have been clearly shown in the indicated **Supplementary Table 5**. Also, it should be noted from the indicated **Supplementary Table 5** that three of four EwS cell lines were in the *PRC1* low group (75%). However, we agree with this Reviewer that this dataset only contains a limited number of EwS xenografts, which is now pointed out more clearly in the text and which is why we have extended our analysis in EwS using as recommended by this Reviewer publicly available DepMap data (see comments on questions #3, #4, and #7, respectively). These analyses showed a strong and significant correlation of *PRC1* expression levels across multiple EwS cell lines with sensitivity toward PLK1 inhibition by Volasertib. These new results have now been embedded in the revised text and are now shown in the **new Supplementary Fig. 4I** and **new Supplementary Table 6** (see comments on questions #3, #4, and #7, respectively).

6. The statement “the rather low PRC1 expression levels and short PRC1-associated GGAA-mSat of the TC71 cell line as demonstrated in the current study” has not been shown, as by mRNA level is very highly expressed in TC71. The authors would need to do a Western of a panel of cell lines to confirm this as I suspect that TC71 will have high expression.

Authors: We politely disagree with this Reviewer. The sentence of our manuscript cited by this Reviewer was followed by a parenthesis, which was omitted in his/her literal citation above, which contained a clear reference to **Supplementary Fig. 2b** in which the mentioned data were and still are clearly shown.

In fact, the initial **Supplementary Fig. 2b** demonstrated a panel of three EwS cell lines among which TC71 exhibited the lowest and A673 the highest *PRC1* mRNA expression levels. However, we agree with this Reviewer that validation of this finding at the protein level will further support our results. Hence, we have analyzed publicly available proteomics data from the DepMap project comprising 4 EwS cell lines including TC71 and A673. Again, we found that TC71 exhibited the lowest and A673 the highest protein expression levels of PRC1. These new data are now shown in the **new Supplementary Fig. 2c**. To further validate the relatively low PRC1 expression in TC71 cells at the protein level, we performed, as suggested by this Reviewer, a western blot analysis of our original three cell lines, which again demonstrated a low PRC1 protein expression in TC71 cells relative to the other EwS cell lines tested. These new results have been integrated in the previous **Supplementary Fig. 2b**. To avoid further confusion, we have copied this Figure with the integrated new western blot data below.

Revised Supplementary Fig. 2b.

New Supplementary Fig. 2c.

Page 5, paragraph 1: “Notably, the average number of GGAA-repeats at this mSat corresponded to the PRC1 expression levels across EwS cell lines at both the mRNA and protein levels

(**Supplementary Fig. 2b**). Consistently, analysis of publicly available proteomics data from the DepMap project (<https://depmap.org/portal/>) comprising 4 EwS cell lines including TC71 and A673 showed that TC71 exhibited the lowest and A673 the highest PRC1 protein levels (**Supplementary Fig. 2c**).”

Also, we have modified the Methods section accordingly as below:

Page 26, paragraph 2: **Western blotting**

“To test for relative PRC1 protein expression levels across EwS cell lines, EwS wt cells were cultured in standard culture condition until reaching 70% confluence.”

Page 39, paragraph 2: **Data and code availability**

“The relative PRC1 protein expression data in EwS cell lines were extracted from the Proteomics (O43663; data censoring 25th May 2021) dataset derived from DepMap portal (<https://depmap.org/portal/>).”

7. The authors could look at much of the drug data e.g. NCI60, Broad data to confirm that the sensitivity to PLK1 is correlated with expression or PRC1.

Authors: We thank this Reviewer for this helpful remark. Unfortunately, the NCI60 data does not contain EwS cell lines. Yet, the DepMap data from the Broad institute comprises a panel of 11 EwS cell lines for which matched *PRC1* mRNA expression data as well as drug sensitivity data for the PLK1 inhibitor BI6727 (Volasertib) were available. Given our previous results from our extensive functional *in vitro* and *in vivo* experiments using the same PLK1 inhibitor, we hypothesized that a higher *PRC1* expression would correlate with a higher sensitivity (that is lower cell viability) toward PLK1 inhibition by Volasertib. Strikingly, despite the still rather low number of EwS cell lines tested, we observed a rather strong negative correlation ($r_{Pearson} = -0.54$) of *PRC1* mRNA levels with lower cell viability upon Volasertib treatment ($P=0.04$, one-sided testing). Since, we have shown in our manuscript that *PRC1* is a direct target gene of the major fusion transcription factor EWSR1-FLI1, present in 85% of EwS cases, we repeated this correlation analysis now only focusing on the 9 of 11 EwS cell lines that exhibited a confirmed *EWSR1-FLI1* fusion according to the DepMap data portal. Even more strikingly,

despite this decrease in sample size, the observed effect size became even stronger (now $r_{Pearson} = -0.72$) and more significant ($P=0.02$, one-sided testing). Importantly, and similar to our observations made in the *in vivo* dataset (see below and reply to question #3 above), such significant correlations were neither observed for *PLK1* nor *MKI67* regardless of the *EWSR1-FLII* status (*PLK1*: $r_{Pearson} = -0.19/-0.40$, $P=0.29/0.14$; *MKI67*: $r_{Pearson} = 0.04/-0.09$, $P=0.46/0.41$ one-sided testing). These new data are now integrated in the revised text and shown in the **new Supplementary Table 6** and **new Supplementary Fig. 4I**. The Methods section and Figure legend have been revised accordingly:

Page 10, paragraph 1: “Together, these findings suggested that genomically silent pediatric cancers, such as EwS, may be very sensitive to PLK1 inhibition in case of high *PRCI* expression. In support of this notion, analysis of matched *in vivo* gene expression and drug-response data from pediatric tumor types (including EwS) (Gorlick *et al.* 2014 Ped Blood Cancer) with relatively silent genomes revealed that good responses to BI6727 (Volasertib) were observed exclusively among *PRCI* high expressing xenografts (defined by median expression; $P=0.0325$, Fisher’s exact test) – an effect not observed for *PLK1* and *MKI67* (**Supplementary Table 5**). However, since this dataset contained xenografts from only 4 different EwS cell lines, we extended our analyses using publicly available drug-response and gene expression data from the DepMap project, comprising 11 EwS cell lines (<https://depmap.org>). In this dataset, we observed a relatively strong negative correlation ($r_{Pearson} = -0.54$) of *PRCI* mRNA levels with lower cell viability upon volasertib treatment ($P=0.04$) (**Supplementary Table 6**). This correlation even remained significant when focusing on the 9 of 11 EwS cell lines that exhibited a confirmed *EWSR1-FLII* fusion ($r_{Pearson} = -0.72$, $P=0.02$) (**Supplementary Fig. 4I**). Similar to our observations made in the *in vivo* dataset (**Supplementary Table 5**) (Gorlick *et al.* 2014 Ped Blood Cancer), such correlations were neither observed for *PLK1* nor *MKI67* regardless of the *EWSR1-FLII* status (*PLK1*: $r_{Pearson} = -0.19/-0.4$, $P=0.29/0.14$; *MKI67*: $r_{Pearson} = 0.04/-0.09$, $P=0.46/0.41$) (**Supplementary Table 6**).”

Page 39, Paragraph 2: **Data and code availability**

“The *PRCI*, *PLK1*, and *MKI67* mRNA expressions of 11 EwS cell lines were extracted from Expression 21Q public dataset and their corresponding drug sensitivities toward the PLK1 inhibitor BI6727 (Volasertib) treatment were extracted from the PRSIM Repurposing Primary Screen 19Q4 derived from DepMap portal (<https://depmap.org/portal/>).”

Reviewers' Comments:

Reviewer #1:

Remarks to the Author:

The authors are to be commended for addressing all of my concern. This manuscript is a tour de force and is a significant contribution to research into Ewing's sarcoma.